# A unified method to revoke the private data of patients in intelligent healthcare with audit to forget

Juexiao Zhou[1,2,3], Haoyang Li [1,2,3], Xingyu Liao [1,2], Bin Zhang[1,2], Wenjia He[1,2], Zhongxiao Li [1,2], Longxi Zhou [1,2] & Xin Gao [1,2] ✉

Revoking personal private data is one of the basic human rights. However, such right is often overlooked or infringed upon due to the increasing collection and use of patient data for model training. In order to secure patients' right to be forgotten, we proposed a solution by using auditing to guide the forgetting process, where auditing means determining whether a dataset has been used to train the model and forgetting requires the information of a query dataset to be forgotten from the target model. We unified these two tasks by introducing an approach called knowledge purification. To implement our solution, we developed an audit to forget software (AFS), which is able to evaluate and revoke patients' private data from pre-trained deep learning models. Here, we show the usability of AFS and its application potential in real-world intelligent healthcare to enhance privacy protection and data revocation rights.

Revoking personal private data is one of the basic human rights, which has already been sheltered by privacy-preserving regulations like The General Data Protection Regulation (GDPR)[1], The Health Insurance Portability and Accountability Act of 1996 (HIPAA)[2], and the California Consumer Privacy Act[3] since twentieth century. With those regulations, users are allowed to request the deletion of their own data for privacy concerns and to secure their own "right to be forgotten". However, with the development of data science, machine learning (ML) and deep learning (DL) techniques, this basic right is usually neglected or violated. For example, it has been observed that patients' genetic markers were leaked from ML methods for genetic data processing[4,5] while the patients were unaware of that. When users realize the existence of such risks, they may request their own data to be deleted to protect their privacy[6]. Meanwhile, those aforementioned regulations will force involved third parties to take actions immediately. According to the requirements of those regulations, not only the previously authorized data by individuals need to be deleted immediately from hosts' storage systems but also the

associated information should be removed from DL models trained with those data, because DL models could memorize sensitive information of training data and thus expose individual's privacy under risk[7–11].

Nowadays, healthcare is one of the most promising areas for the deployment of artificial intelligent (AI) systems as so-called intelligent healthcare. ML and DL-based computer-aided diagnosis (CAD) systems in intelligent healthcare accelerate the diagnosis of various diseases and achieve even better results than doctors, such as tumor detection[12,13], retinal fundus imaging[14], detection and segmentation of COVID-19 lung infections[15,16] and so on. However, as more and more patients' data are being collected and used for model training in intelligent healthcare, their privacy is exposed to high risk. Therefore, intelligent healthcare is a sector where technology must meet the law, regulations, and privacy principles to ensure that the innovation is for the common good[17]. To obey those privacy-preserving regulations, methods to revoke personal private data from pre-trained DL models are necessary.

[1]Computer Science Program, Computer, Electrical and Mathematical Sciences and Engineering Division, King Abdullah University of Science and Technology (KAUST), 23955-6900 Thuwal, Kingdom of Saudi Arabia. [2]Computational Bioscience Research Center, Computer, Electrical and Mathematical Sciences and Engineering Division, King Abdullah University of Science and Technology (KAUST), 23955-6900 Thuwal, Kingdom of Saudi Arabia. [3]These authors contributed equally: Juexiao Zhou, Haoyang Li. ✉e-mail: xin.gao@kaust.edu.sa

Deleting the stored personal data is simple, whereas forgetting individuals' private information from pre-trained DL models could be difficult as we could not fully measure the contribution of individual data on the training process of DL models due to the stochasticity of training[18]. Besides, due to the incremental nature of training, the model update brought by one sample would affect the model performance on samples followed, thus making it difficult to unlearn[18]. Finally, catastrophic unlearning might happen and the unlearned model will perform worse than the model retrained on the remaining dataset[19].

In general, the process to forget data from a pre-trained DL model could be divided into two steps. Firstly, the unlearning process (forgetting) is performed on a given pre-trained DL model to forget the target data with different techniques and a new DL model will be generated. Secondly, an evaluation of the new model (auditing) against different metrics will be performed to prove that the model has forgotten the target data. These two processes should be repeated until the new model passes the evaluation. In simple terms, there are two commonly acknowledged sub-tasks, which could also be stated in the reverse order: auditing and forgetting, as a two-player game. Auditing requires auditors to precisely evaluate whether the data of certain patients were used to train the target DL model. Once the data of certain patients is confirmed to be used to train the target DL model by auditing, forgetting requires the removal of learnt information of certain patients' data from the target DL model, which is also called machine unlearning, while auditing could act as the verification of machine unlearning[18].

In order to achieve forgetting, existing unlearning methods could be classified into three major classes, including model-agnostic methods, model-intrinsic methods and data-driven methods[20]. Model-agnostic methods refer to algorithms or frameworks that can be used for different DL models, including differential privacy[18,21–23], certified removal[24–26], statistical query learning[6], decremental learning[27], knowledge adaptation[28,29] and parameter sampling[30]. Model-intrinsic approaches are those methods designed for specific types of models, such as for softmax classifiers[31], linear models[32], tree-based models[33] and Bayesian models[19]. Data-driven approaches focus on the data itself, including data partitioning[18], data augmentation[34–36] and other unlearning strategies based on data influence[37]. However, most of them are theoretical studies and do not provide open-source codes. Besides, most methods have their specific application scenarios and acknowledged limitations and few of them focused on the application in real-world intelligent healthcare. Among the three methods, model-agnostic methods might have the strongest application prospects, as they can be applied to different models, and SISA, short for

Sharded, Isolated, Sliced, and Aggregated training, is the most classic and well-known method in the community[38]. As the state-of-the-art method, Goel et al.[39] proposed Catastrophic Forgetting-k (CF-k) and Exact Unlearning-k (EU-k) to unlearn information from deep learning models. CF-k means to finetune the last k layers of the original model on $D_r$ and freeze other layers, while EU-k means to retrain the last k layers of the original model from scratch on $D_r$ and freeze other layers, where the $D_r$ stands for the retain dataset.

When forgetting is accomplished, auditing is the next necessary step to verify it. Different metrics have been proposed to audit the membership of the query dataset, including accuracy, completeness[6], unlearn time, relearn time, retrain time, layer-wise distance, activation distance, JS-divergence, membership inference[40,41], ZRF score[28], epistemic uncertainty[42] and model inversion attack[7]. In recent studies, membership inference-based metrics were frequently utilized to determine whether or not any information about the samples to be forgotten was retained in the model in intelligent healthcare[41]. A black-box setting was shared by the membership inference attack (MIA) to calculate the probability of a single datapoint being a member of the training dataset $D$. Based on this individual-level MIA, Liu et al.[40] and Yangsibo et al.[41] focused on a more challenging task: audit the membership of a set of data points. The ensembled membership auditing (EMA)[41] was proposed as the state-of-the-art method to verify whether a query dataset is memorized by a pre-trained DL model, which is also a benchmark metric in machine unlearning. However, due to the black box property of DL models, efficient and accurate auditing is still challenging and an under-studied topic. Moreover, researchers have tended to treat auditing and forgetting as separate tasks, ignoring the fact that the two can be linked up associatively to work as a self-consistent mechanism.

Here, we proposed a solution by using auditing to guide the forgetting process in a negative feedback manner. We unified the two tasks by introducing knowledge purification (KP), an approach to selectively transfer the needed knowledge to forget the target information instead of simply transferring all information like knowledge distillation (KD)[43]. On the basis of KP, we have developed a user-friendly and open-source audit to forget software (AFS) (Fig. 1), which can be easily used to revoke patients' private data from DL models in intelligent healthcare with KP (Fig. 2). To demonstrate the generality of AFS, we applied it to four tasks based on four datasets, including the MNIST dataset, the PathMNIST dataset, the COVIDx dataset, and the ASD dataset, with different data sizes and various architectures of deep learning networks (Fig. 3). Our results demonstrate the usability of AFS and its application potential in real-world intelligent healthcare to

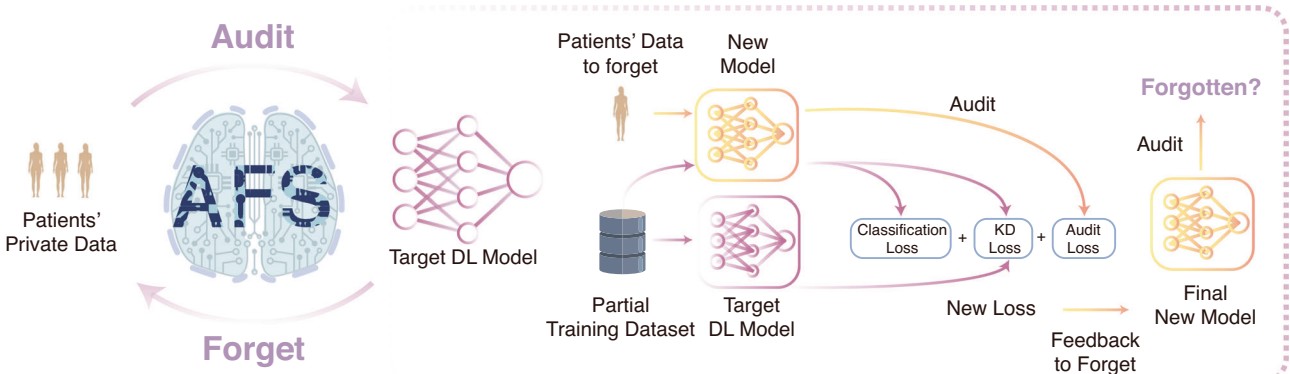

**Fig. 1 | AFS is a unified method to revoke patients' private data in intelligent healthcare.** The left side of the figure illustrates the high-level iterative flow of AFS, while the right side illustrates the details of how forgetting and auditing work together. As shown on the left side, given a pre-trained DL model and a query dataset (patients' private data), AFS could audit and provide confidence whether the query dataset has been used to train the target DL model. When a dataset has been used to train the target DL model, AFS could effectively forget the information about the dataset from the target DL model with the guidance of auditing. To achieve that, we proposed a method called knowledge purification as shown on the right side, which utilizes results from auditing as a new term in the loss function to forget information. The brain icon is Designed by macrovector/Freepik.

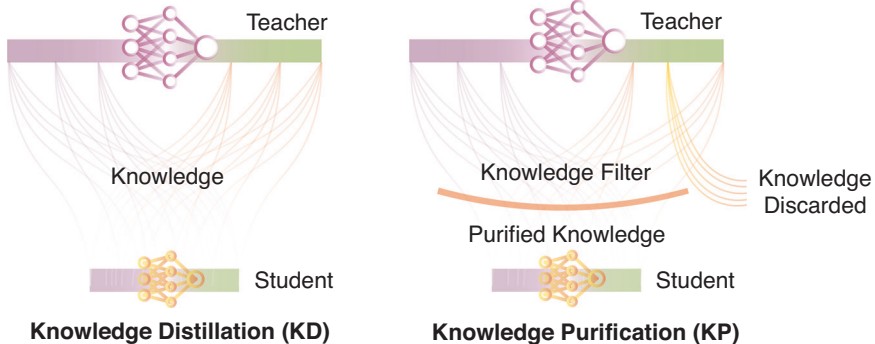

**Fig. 2 | Illustration of knowledge distillation and knowledge purification.** Knowledge purification requires the selective transfer of the needed knowledge in the process of knowledge distillation to forget the target information instead of simply transferring all information.

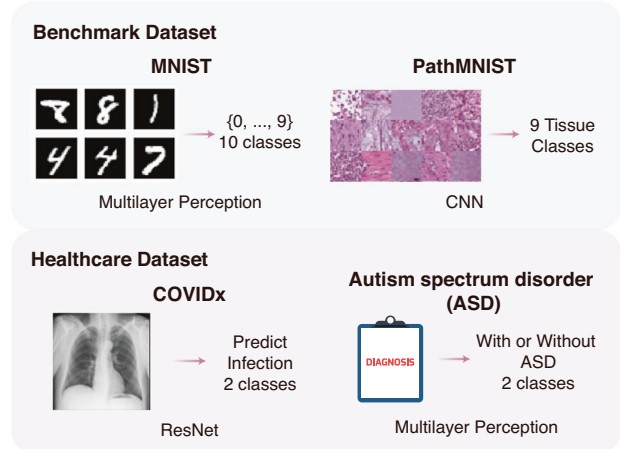

**Fig. 3 | Illustration of four datasets and DL models used to show the versatility of AFS.** Two benchmark datasets (MNIST and PathMNIST) and two healthcare datasets (COVIDx and Autism spectrum disorder) were used in this work.

enhance privacy protection and data revocation rights. AFS is a unified method of auditing and forgetting that could effectively forget the information of the target query dataset from the pre-trained DL model with the guidance of auditing. AFS could generate a smaller model, which requires much less time and GPU memory during the inference, by training with a partial training dataset (~50%) with our KP approach.

## Results

### AFS audits private datasets stably and robustly

To evaluate the robustness of auditing by AFS, we used it to audit query datasets with different sizes, various purity (k percent of the query dataset was overlapped with the training dataset) and the different sizes of calibration dataset (the size ranged from 100 to 5000) ("Method" and Fig. 4a). For each sample in the query dataset, AFS calculates three metrics for the membership inference, including correctness, confidence and negative entropy ("Method"). As shown in Figs. 4b and S1, all three metrics showed different distributions for QO (query dataset overlapped with the training dataset) and QNO (query dataset disjoint with the training dataset), indicating the dataset-wise divergence of metrics between samples in the training dataset and samples disjoint with the training dataset. Given the differences observed in the three metrics, we could distinguish between QO and QNO by calculating the $p$ value and utilizing it as the sole audit metric for forgetting (Method). Finally, by integrating these three metrics, AFS reports a $p$ value to evaluate whether or not a query dataset has been used to train the target DL model. The large $p$ values indicate the higher probability that the query dataset was used in training.

When the size of the query dataset and the calibration dataset varied, AFS could still efficiently distinguish QO and QNO (Fig. 4c, d). Compared to QO, AFS reported a much smaller $p$ value for QNO, indicating a weak membership (a small probability that the query dataset has been used to train the target DL model), thus allowing users to judge whether the query dataset was used to train the target DL model. Meanwhile, when the size of the dataset increased from 1 to 2000, AFS discriminated QO and QNO more confidently as there was a more significant divergence of the $p$ values, which was not affected by the size of the calibration dataset. To further understand the effect of the purity of the query dataset in auditing, we mixed some samples from the training dataset to QNO, thus the new query dataset was labeled as QM (partial data overlapped with the training dataset). The percentage of data overlapped with the training dataset in QM was denoted by $k = \frac{\text{number of data overlapped with training dataset}}{\text{size of QM}}$. As shown in Fig. 4e, AFS showed a decreasing $p$ value trend when $k$ decreased, meaning that the query dataset was less likely to be used to train the target DL model. The distinctiveness of the $p$ value on the ASD dataset is due to the small size of the ASD dataset. $k = 0$ was shown in Fig. 4d as QNO, which occurred when the query dataset size was 100 for ASD and 2000 for MNIST, PathMNIST, and COVIDx. Similarly, $k = 1$ was also displayed in Fig. 4d as QO, which occurred when the query dataset size was 100 for ASD and 2000 for MNIST, PathMNIST, and COVIDx. In conclusion, these results indicate the robustness of AFS in determining whether the query data has been used to train the target DL model.

### AFS forgets the information of query dataset, maintains perfect usability and generates smaller model

Once the prior knowledge that a dataset has been used to train the target DL model is confirmed with auditing, AFS could be used for forgetting, to remove the information of the dataset from the pre-trained DL model. To comprehensively show the ability of AFS in removing information against the model performance, we compared nine methods, including (1) training the teacher model with a complete training dataset (Independent teacher), (2) retraining the student model with a complete training dataset (Independent student), (3) retraining the teacher model with $k \in \{0.25, 0.5, 0.75\}$ percentage of the complete training dataset excluding the data to be forgotten (Independent teacher with $k \in \{0.25, 0.5, 0.75\}$), (4) retraining the student model with $k \in \{0.25, 0.5, 0.75\}$ percentage of the complete training dataset excluding the data to be forgotten (Independent Student with $k \in \{0.25, 0.5, 0.75\}$), (5) retraining the model of the corresponding shard with SISA, (6) fine-tuning the last layers of the model with CF-k, (7) retraining the last layers of the model from scratch with EU-k, (8) AFS, and (9) training the student model with AFS without the guidance of auditing (AFS w/o Audit), as an ablation study of AFS. Both AFS w/o Audit and AFS were also conducted with varied $k \in \{0.25, 0.5, 0.75\}$. For both Independent

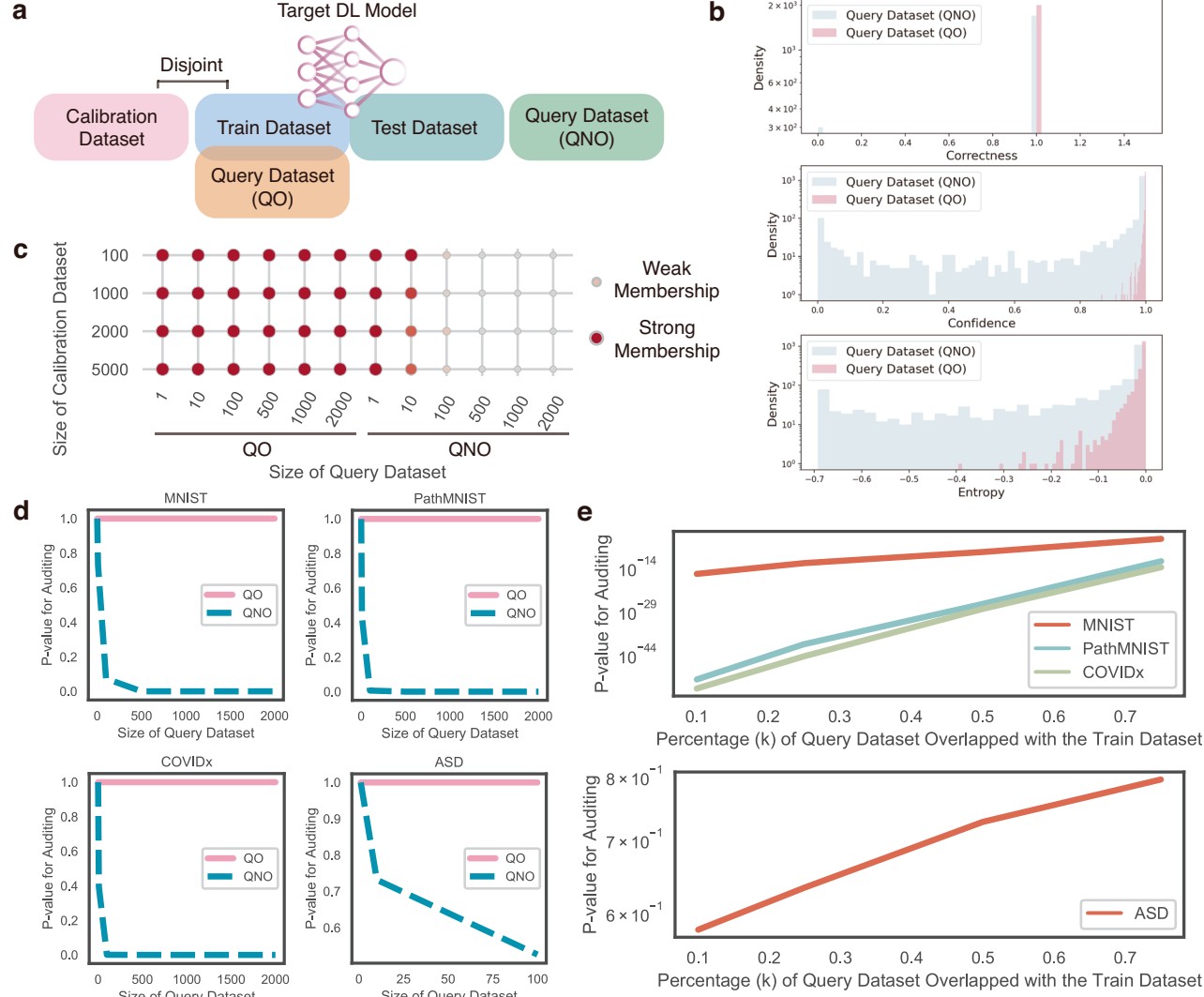

**Fig. 4 | Performance of auditing using AFS on the four datasets.**
**a** Demonstration of the training dataset, the test dataset, the calibration dataset, and the query dataset overlapped with the training dataset (QO) and the query dataset disjointed with the training dataset (QNO). **b** Distribution of three metrics for samples in QO and QNO. **c** The performance of auditing when varying the size of the calibration dataset and the size of the query dataset. **d** The *p* value of auditing on QO and QNO of four datasets. *p* values were calculated using two-tailed Student's *t* test. **e** The *p* value of auditing when varying *k* of the query dataset of four datasets. *p* values were calculated using two-tailed Student's *t* test.

teacher and Independent student methods trained with the complete training dataset, QF$_{100}$ and QF$_{1000}$ were included in the training dataset, while these two query datasets were excluded from the training dataset when $k \in \{0.25, 0.5, 0.75\}$.

Taking the MNIST dataset as an example, for models trained with each method, except for auditing on QO and QNO, we further audited the membership of two datasets designed to be forgotten (a small query dataset QF$_{100}$ and a large query dataset QF$_{1000}$) to assess the ability of different methods in forgetting the query dataset. As shown in Table 1, regardless of the model trained based on which method, AFS could effectively distinguish between QO and QNO, and the divergence in auditing two query datasets was enlarged as the size of the query dataset increased.

As shown in Table 2, AFS perfectly predicted the membership of QF$_{100}$ and QF$_{1000}$ on both models from Independent teacher and Independent student methods as both query datasets were included in the training dataset. Since both query datasets were disjoint with the partial training dataset when $k \in \{0.25, 0.5, 0.75\}$, thus auditing on the model trained with Independent teacher and Independent student with $k \in \{0.25, 0.5, 0.75\}$ weakly denied the membership of QF$_{100}$

($P_{QF100,k=0.75} = 1.57E - 1$, $P_{QF100,k=0.5} = 1.56E - 1$, $P_{QF100,k=0.25} = 8.17E - 2$ for Independent teacher and $P_{QF100,k=0.75} = 4.36E - 2$, $P_{QF100,k=0.5} = 6.91E - 3$, $P_{QF100,k=0.25} = 6.91E - 3$ for Independent student) and QF1000 ($P_{QF1000,k=0.75} = 1.05E - 8$, $P_{QF1000,k=0.5} = 2.71E - 11$, $P_{QF1000,k=0.25} = 2.80E - 18$ for Independent teacher and $P_{QF1000,k=0.75} = 5.26E - 12$, $P_{QF1000,k=0.5} = 2.34E - 15$, $P_{QF1000,k=0.25} = 2.90E - 19$ for Independent student). However, since only the partial training dataset was used when $k \in \{0.25, 0.5, 0.75\}$, the retrained models with Independent student and Independent student only learnt the information of the partial training dataset and lost the information from the remaining data in the complete training dataset, thus resulting in the significant drop of model performance compared to either the Independent student or the Independent teacher trained with the complete training dataset. The same conclusion could be drawn with SISA (The training data were divided into 10 shards. Then, QF was removed from the shard where the QF was located, and the model of the corresponding shard was re-trained to re-aggregate the final model). Meanwhile, due to the unique design, SISA requires storing 10 model parameters simultaneously, resulting in greater storage consumption than AFS. Although CF-k and EU-k achieved good accuracy and F1-score, the

**Table 1 | Comparison of AFS with other methods on auditing QO and QNO from the MNIST dataset with a varied number of samples in the query dataset**

| Methods | QO | | | | | | QNO | | | | | |
|---|---|---|---|---|---|---|---|---|---|---|---|---|
| | 1 | 10 | 100 | 500 | 1000 | 2000 | 1 | 10 | 100 | 500 | 1000 | 2000 |
| Independent teacher | 1 | 1 | 1 | 1 | 1 | 1 | 1 | 7.32e−01 | 7.39e−02 | 2.43e−05 | 1.06e−08 | 4.68e−18 |
| Independent student | 1 | 1 | 1 | 3.43e−01 | 4.22e−01 | 8.61e−02 | 1 | 6.96e−01 | 4.50e−02 | 2.74e−06 | 4.51e−11 | 8.58e−22 |
| Independent teacher ($k = 0.75$) | 1 | 1 | 1 | 1 | 1 | 1 | 1 | 5.62e−01 | 8.84e−02 | 2.38e−05 | 3.93e−09 | 1.87e−19 |
| Independent student ($k = 0.75$) | 1 | 1 | 1 | 6.80e−01 | 1.67e−01 | 7.87e−02 | 1 | 5.27e−01 | 2.28e−02 | 7.50e−07 | 4.61e−14 | 3.10e−27 |
| AFS w/o Audit ($k = 0.75$) | 1 | 1 | 1 | 8.64e−01 | 5.91e−01 | 5.26e−01 | 1 | 5.98e−01 | 1.02e−01 | 6.49e−06 | 4.15e−11 | 1.28e−20 |
| AFS ($k = 0.75$) | 1 | 1 | 8.64e−01 | 1 | 8.64e−01 | 4.54e−01 | 1 | 6.96e−01 | 2.94e−02 | 1.45e−06 | 9.06e−13 | 3.79e−24 |
| Independent teacher ($k = 0.5$) | 1 | 1 | 1 | 1 | 1 | 1 | 1 | 6.96e−01 | 1.56e−02 | 1.84e−06 | 1.67e−12 | 4.81e−26 |
| Independent student ($k = 0.5$) | 1 | 1 | 8.64e−01 | 5.59e−01 | 2.39e−01 | 2.49e−02 | 1 | 8.13e−01 | 7.75e−02 | 6.44e−08 | 1.63e−15 | 4.15e−30 |
| AFS w/o Audit ($k = 0.5$) | 1 | 1 | 1 | 8.64e−01 | 8.64e−01 | 8.64e−01 | 1 | 5.98e−01 | 1.32e−01 | 5.94e−06 | 2.43e−12 | 5.73e−23 |
| AFS ($k = 0.5$) | 1 | 1 | 1 | 8.64e−01 | 1 | 7.27e−01 | 1 | 6.68e−01 | 2.88e−02 | 9.25e−07 | 3.70e−13 | 3.10e−27 |
| Independent teacher ($k = 0.25$) | 1 | 1 | 1 | 1 | 1 | 1 | 1 | 8.66e−01 | 5.27e−02 | 6.05e−06 | 3.24e−18 | 9.75e−34 |
| Independent student ($k = 0.25$) | 1 | 1 | 1 | 8.64e−01 | 8.64e−01 | 3.17e−01 | 1 | 5.98e−01 | 1.04e−02 | 6.40e−10 | 3.05e−20 | 1.22e−38 |
| AFS w/o Audit ($k = 0.25$) | 1 | 1 | 1 | 1 | 1 | 1 | 1 | 5.27e−01 | 4.47e−02 | 7.49e−07 | 2.18e−13 | 5.96e−28 |
| AFS ($k = 0.25$) | 1 | 1 | 1 | 1 | 1 | 1 | 1 | 6.96e−01 | 8.52e−03 | 3.85e−10 | 1.98e−20 | 1.29e−41 |
| SISA (10 shards) | 1 | 1 | 1 | 1 | 1 | 1 | 1 | 7.08e−01 | 7.20e−02 | 4.61e−05 | 3.81e−09 | 6.34e−21 |
| CF-k ($k = 1$) | 1 | 1 | 1 | 1 | 1 | 1 | 1 | 7.32e−01 | 1.11e−01 | 7.64e−05 | 5.48e−08 | 2.32e−17 |
| EU-k ($k = 1$) | 1 | 1 | 1 | 8.66e−01 | 1 | 4.91e−01 | 1 | 5.98e−01 | 4.10e−02 | 1.29e−05 | 7.96e−11 | 2.90e−22 |

The data in the table show the results of auditing QO and QNO on models trained by different methods. The numerical values in the table represent *p* values. A larger value indicates stronger membership. *p* values were calculated using two-tailed Student's *t* test.

**Table 2 | Comparison of AFS with other methods on forgetting QF and model performance with the MNIST dataset**

| Methods | $QF_{100}$ | $QF_{1000}$ | Accuracy | F1-score |
|---|---|---|---|---|
| Independent teacher | 1 | 1 | 0.9622 | 0.9911 |
| Independent student | 1 | 1 | 0.9504 | 0.9911 |
| Independent teacher ($k = 0.75$) | 1.57e−01 | 1.05e−08 | 0.9548 | 0.9915 |
| Independent student ($k = 0.75$) | 4.36e−02 | 5.26e−12 | 0.9458 | 0.9880 |
| AFS w/o Audit ($k = 0.75$) | 3.19e−01 | 1.33e−06 | 0.9582 | 0.9884 |
| AFS ($k = 0.75$) | 1.08e−03 | 5.22e−23 | 0.9470 | 0.9889 |
| Independent teacher ($k = 0.5$) | 1.56e−01 | 2.71e−11 | 0.9437 | 0.9898 |
| Independent student ($k = 0.5$) | 6.91e−03 | 2.34e−15 | 0.9282 | 0.9848 |
| AFS w/o Audit ($k = 0.5$) | 1.57e−01 | 7.79e−07 | 0.9526 | 0.9884 |
| AFS ($k = 0.5$) | 1.27e−02 | 9.44e−22 | 0.9380 | 0.9866 |
| Independent teacher ($k = 0.25$) | 8.17e−02 | 2.80e−18 | 0.9176 | 0.9871 |
| Independent student ($k = 0.25$) | 6.91e−03 | 2.90e−19 | 0.9067 | 0.9875 |
| AFS w/o Audit ($k = 0.25$) | 1.57e−01 | 7.04e−10 | 0.9388 | 0.9875 |
| AFS ($k = 0.25$) | 5.79e−04 | 6.84e−33 | 0.9174 | 0.9853 |
| SISA (10 shards, QF excluded, $k = 0.99$ for $QF_{100}$ and 0.9 for $QF_{1000}$ | 3.18e−01 | 6.12e−09 | 0.9568 | 0.9911 |
| CF-k (QF excluded) | 1 | 1 | 0.9608 | 0.9907 |
| EU-k (QF excluded) | 1 | 1.41e−02 | 0.9504 | 0.9898 |

QF100 is the small query dataset containing 100 samples and QF1000 is the large query dataset containing 1000 samples. We present the *p* values of auditing models trained with different methods on QF100 and QF1000 and the model performance including the accuracy and F1-score. *p* values were calculated using two-tailed Student's *t* test.

audit results showed that only fine-tuning or retraining the last few layers of the original model is not enough to forget the information of the query data.

To rescue the information lost due to the usage of partial training samples and further increase the model performance, AFS could use only a partial training dataset ($k \in \{0.25, 0.5, 0.75\}$) to transfer the knowledge from the Independent teacher pre-trained with the complete training dataset. As shown in Table 2, the model trained with AFS provided higher accuracy and F1-score compared to the Independent student trained with partial training dataset ($k \in \{0.25, 0.5, 0.75\}$) and together with a better forgetting performance (much smaller auditing score on $QF_{100}$ and $QF_{1000}$), as AFS used auditing as feedback for forgetting and could forget not only the query samples but also other samples with similar features.

We also applied AFS on the 9-classes classification of hematoxylin and eosin-stained histological images from the PathMNIST dataset with CNN. As shown in Table 3, AFS could still distinguish QO and QNO from the PathMNIST dataset. The divergence of auditing between QO and QNO was more significant than that on the MNIST dataset. With the requirement to forget both query datasets (QF$_{100}$ and QF$_{1000}$), the model trained with AFS outperformed on forgetting information ($P_{QF100,k=0.75} = 2.25E - 5$, $P_{QF100,k=0.5} = 2.87E - 6$, $P_{QF100,k=0.25} = 3.32E - 7$, $P_{QF1000,k=0.75} = 2.05E - 41$, $P_{QF1000,k=0.5} = 4.75E - 35$, $P_{QF1000,k=0.25} = 1.84E - 56$) while learnt more information from the Independent teacher model trained with a complete training dataset.

In summary, AFS could effectively forget the information of the query dataset from the target DL model. Since KP was integrated into AFS, it could generate a smaller DL model, which masters knowledge from the larger teacher model by using only a partial training dataset ($k = 0.5$ could achieve a good balance between forgetting and model performance), without the need to retrain the larger model with the complete training dataset. Compared to retraining the student model, the model trained with AFS showed even better performance in forgetting the information while maintaining better model performance (accuracy and F1-score) as it learnt the knowledge from the model trained with the complete training dataset. As shown by the ablation study in Tables 2 and 4, compared to AFS w/o Audit, the audit-guided AFS could forget the information more significantly but with an acceptable cost in decreasing the model performance (accuracy and F1-score).

## Apply AFS to forget medical images
To show the versatility of AFS, we applied it to the classification of pneumonia and normal with chest X-ray images from the COVIDx dataset with ResNet, which is a classic task in medical image analysis. As shown in Fig. 5a, on both query datasets (QF$_{100}$ and QF$_{1000}$), AFS could

effectively forget the information of the query dataset, while generating the new model with much less number of parameters as shown in Fig. 5b. Surprisingly, the model generated by AFS showed even better accuracy than the Independent teacher trained with the complete dataset and the Independent student trained with the partial training dataset. This result not only indicated that AFS could effectively transfer the knowledge from the teacher model to the student model but also suggested that the student model with simpler architecture could even perform better than the teacher model with KP in AFS due to the reduction of model parameters and purification of knowledge in some real-world cases. Meanwhile, the results also showed that the student model trained by the AFS achieved better generalization ability than the model obtained by using the exact same model structure and training data using only the hard-target training method.

## Apply AFS to forget electrical health records
To further prove the generalizability of AFS in both the auditing and forgetting, we applied AFS to predicting early autism spectrum disorder (ASD) traits of toddlers, which contains sensitive information about patients, such as the age, gender and the family gene trait. That information was stored as electrical health records (EHR). As shown in Fig. 5a, similar to previous results on other datasets, AFS effectively removed the information of both query datasets from the pre-trained DL model. Since the size of the ASD dataset was quite small, we adopted two smaller query datasets (QF$_{50}$ and QF$_{100}$) to be forgotten. Compared to the models trained with other methods, the model trained with AFS successfully forgot the information of both QF$_{50}$ ($P_{QF50,k=0.75} = 0.08, P_{QF50,k=0.55} = 0.08, P_{QF50,k=0.25} = 0.156$) and QF$_{100}$ ($P_{QF100,k=0.75} = 0.004, P_{QF100,k=0.55} = 0.007, P_{QF100,k=0.25} = 0.007$) without affecting the model utility significantly ($Acc_{AFS,k=0.75} = 0.98, Acc_{AFS,k=0.5} = 0.98, Acc_{AFS,k=0.25} = 0.98$).

**Table 3 | Comparison of AFS with other methods on auditing QO and QNO from the PathMNIST dataset with a varied number of samples in the query dataset**

| Methods | QO | | | | | | QNO | | | | | |
|---|---|---|---|---|---|---|---|---|---|---|---|---|
| | 1 | 10 | 100 | 500 | 1000 | 2000 | 1 | 10 | 100 | 500 | 1000 | 2000 |
| Independent teacher | 1 | 1 | 1 | 1 | 1 | 1 | 1 | 4.29e-01 | 6.29e-03 | 1.40e-13 | 4.91e-28 | 1.88e-60 |
| Independent student | 1 | 1 | 1 | 1 | 1 | 1 | 1 | 5.27e-01 | 5.11e-03 | 1.29e-10 | 3.38e-20 | 7.24e-42 |
| Independent teacher ($k = 0.75$) | 1 | 1 | 1 | 1 | 1 | 1 | 1 | 5.62e-01 | 2.76e-03 | 4.22e-16 | 5.52e-27 | 1.62e-60 |
| Independent student ($k = 0.75$) | 1 | 1 | 1 | 1 | 1 | 1 | 1 | 7.32e-01 | 1.89e-02 | 1.55e-08 | 5.74e-16 | 1.92e-30 |
| AFS w/o Audit ($k = 0.75$) | 1 | 1 | 5.26e-01 | 1.78e-01 | 2.54e-02 | 1.17e-03 | 1 | 2.95e-01 | 2.26e-02 | 9.23e-11 | 1.04e-19 | 8.21e-41 |
| AFS ($k = 0.75$) | 1 | 1 | 3.90e-01 | 3.21e-02 | 6.13e-04 | 7.19e-06 | 1 | 4.98e-01 | 2.34e-04 | 4.25e-16 | 3.95e-31 | 4.89e-65 |
| Independent teacher ($k = 0.5$) | 1 | 1 | 1 | 1 | 1 | 1 | 1 | 5.62e-01 | 5.03e-03 | 1.50e-12 | 3.36e-22 | 2.39e-48 |
| Independent student ($k = 0.5$) | 1 | 8.66e-01 | 1 | 8.64e-01 | 8.64e-01 | 5.58e-01 | 1 | 5.62e-01 | 4.02e-03 | 3.01e-10 | 1.91e-21 | 7.85e-40 |
| AFS w/o Audit ($k = 0.5$) | 1 | 1 | 5.11e-01 | 2.48e-01 | 1.24e-02 | 9.59e-04 | 1 | 6.26e-01 | 1.18e-02 | 2.99e-11 | 4.03e-21 | 2.51e-40 |
| AFS ($k = 0.5$) | 1 | 1 | 1.59e-01 | 3.45e-04 | 4.06e-06 | 1.85e-10 | 1 | 2.69e-01 | 9.17e-05 | 2.98e-16 | 3.84e-30 | 1.04e-62 |
| Independent teacher ($k = 0.25$) | 1 | 1 | 1 | 1 | 1 | 1 | 1 | 3.31e-01 | 5.98e-04 | 2.84e-12 | 1.62e-28 | 6.34e-52 |
| Independent student ($k = 0.25$) | 1 | 1 | 8.64e-01 | 6.95e-01 | 1.60e-01 | 4.54e-02 | 1 | 1.26e-01 | 1.24e-06 | 2.30e-31 | 2.88e-60 | 4.73e-109 |
| AFS w/o Audit ($k = 0.25$) | 1 | 8.66e-01 | 2.39e-01 | 3.37e-03 | 1.01e-05 | 9.15e-10 | 1 | 2.42e-01 | 3.64e-05 | 2.55e-24 | 3.01e-43 | 1.37e-90 |
| AFS ($k = 0.25$) | 1 | 8.66e-01 | 1.40e-01 | 2.71e-03 | 1.07e-06 | 1.63e-12 | 1 | 4.37e-01 | 7.07e-06 | 2.01e-27 | 4.45e-53 | 1.71e-101 |
| SISA (10 shards) | 1 | 1 | 1 | 1 | 1 | 1 | 1 | 6.71e-01 | 6.58e-03 | 2.11e-12 | 9.05e-27 | 5.85e-49 |
| CF-k ($k = 1$) | 1 | 1 | 1 | 8.63e-01 | 8.63e-01 | 1 | 1 | 3.75e-01 | 7.28e-03 | 9.10e-14 | 1.07e-21 | 1.48e-42 |
| EU-k ($k = 1$) | 1 | 1 | 1 | 6.96e-01 | 7.38e-01 | 6.92e-01 | 1 | 4.73e-01 | 1.60e-03 | 4.66e-11 | 1.47e-23 | 1.48e-44 |

The data in the table show the results of auditing QO and QNO on models trained by different methods. The numerical values in the table represent $p$ values. A larger value indicates stronger membership. $p$ values were calculated using two-tailed Student's $t$ test.

**Table 4 | Comparison of AFS with other methods on forgetting QF and model performance with the PathMNIST dataset**

| Methods | $QF_{100}$ | $QF_{1000}$ | Accuracy | F1-score |
|---|---|---|---|---|
| Independent teacher | 1 | 1 | 0.8538 | 0.9885 |
| Independent student | 1 | 1 | 0.8446 | 0.9836 |
| Independent teacher ($k = 0.75$) | 1.08e−03 | 3.11e−30 | 0.8214 | 0.9796 |
| Independent student ($k = 0.75$) | 2.35e−02 | 4.08e−15 | 0.8396 | 0.9555 |
| AFS w/o Audit ($k = 0.75$) | 1.08e−03 | 1.67e−22 | 0.8682 | 0.9777 |
| AFS ($k = 0.75$) | 2.25e−05 | 2.05e−41 | 0.8560 | 0.9605 |
| Independent teacher ($k = 0.5$) | 3.74e−03 | 2.91e−23 | 0.7100 | 0.8314 |
| Independent student ($k = 0.5$) | 6.91e−03 | 2.99e−21 | 0.7934 | 0.9533 |
| AFS w/o Audit ($k = 0.5$) | 3.74e−03 | 4.93e−18 | 0.8494 | 0.9697 |
| AFS ($k = 0.5$) | 2.87e−06 | 4.75e−35 | 0.8242 | 0.9575 |
| Independent teacher ($k = 0.25$) | 3.74e−03 | 2.52e−26 | 0.7026 | 0.8282 |
| Independent student ($k = 0.25$) | 1.58e−07 | 9.05e−57 | 0.7582 | 0.9287 |
| AFS w/o Audit ($k = 0.25$) | 3.32e−07 | 2.05e−41 | 0.7842 | 0.9406 |
| AFS ($k = 0.25$) | 3.32e−07 | 1.84e−56 | 0.7810 | 0.9385 |
| SISA (10 shards, QF excluded, $k = 0.99$ for $QF_{100}$ and 0.9 for $QF_{1000}$) | 2.15e−03 | 6.73e−29 | 0.8501 | 0.9840 |
| CF-k (QF excluded) | 1 | 1.69e−02 | 0.8506 | 0.9839 |
| EU-k (QF excluded) | 2.35e−02 | 7.69e−16 | 0.8328 | 0.9596 |

QF100 is the small query dataset containing 100 samples and QF1000 is the large query dataset containing 1000 samples. We present the *p* values of auditing models trained with different methods on QF100 and QF1000 and the model performance including the accuracy and F1-score. *p* values were calculated using two-tailed Student's *t* test.

## Discussion

AFS is a unified method of auditing and forgetting that could effectively forget the information of the target query dataset from the pretrained DL model with the guidance of auditing. We designed AFS as a model-agnostic and open-source method that is applicable to different models. As shown in Fig. 5c, AFS could generate a smaller model, which requires much less time and GPU memory during the inference (Tables S1 and S2), by training with a partial training dataset (~50%) with our KP approach. Moreover, AFS could forget the information of the query dataset at the expense of an acceptable reduction in the model performance.

Our experiments on four datasets showed that AFS was generalized for datasets of different sizes and forms, including medical images and EHR. Since deep learning models with different architectures were applied to four tasks, we further demonstrated the broad applicability of AFS to common deep learning models. In addition, our tasks include both binary classification and multiclassification tasks, which also suggested that AFS was applicable for tasks with multiple labels.

In practice, the size of the student model could be manually adjusted when applying AFS to meet specific requirements. In the initial stages, we could make the student model smaller than the teacher model to achieve model compression while forgetting private information. However, instead of continuously generating smaller and smaller student models as knowledge is forgotten, we could maintain the size of the student model once it reaches a certain threshold (e.g., small enough to meet our needs). In doing so, AFS would focus solely on forgetting knowledge and transferring the remaining knowledge, without the need for further model compression.

AFS could be incorporated into the workflow of institutions as shown in Fig. 5d. Patients' requests for data forgetting may occur in two different phases. The first phase relates to requests made before model compression, where patients request the institution to forget their data from the initial dataset. The second phase pertains to requests made after model compression, where patients seek to have their data forgotten from the compressed model. In both cases, AFS could be employed to forget information from the model. In the first case, AFS facilitates model compression while performing forgetting, which ensures that the compressed model not only meets the requirements for deployment but also respects data privacy by

removing sensitive information. In the second case, AFS could be utilized to forget information from the model without involving model compression. This allows the institution to respond to data forget requests while retaining the compressed model's structure and size. The institution may stop forgetting and retrain a new model in two possible scenarios. The first scenario occurs when the number of forgetting requests becomes too high, leading to a significant degradation in the current model's performance that exceeds the institution's predefined budget. In such cases, it may be necessary to stop the forgetting process and initiate the retraining of a new model. The second scenario arises when the institution introduces new data into the system. In this case, instead of continuing to forget specific records from the existing model, the institution can incorporate the retained data along with the new data to train a new model.

With current laws that guarantee people the right to revoke their own data, AFS could help institutions and companies to efficiently iterate their models to forget individual information at the model level. However, there are still some shortcomings in the application of the current version of AFS in the production environment, which could be the main potential direction of research in the future. Firstly, the models and data we tested in this study were still not large enough compared to the data in the real production environment. Therefore, it is unknown whether scaling AFS to current large models will cause new problems (e.g. LLM like ChatGPT, which we are unable to further pursue). Secondly, there are different approaches to audit, and thus we could add more metrics of auditing to AFS to guide the forgetting process in the future version. Meanwhile, due to the limitation of auditing, it is still difficult to perform individual-level forgetting, as we need to compare the difference in statistical distribution based on a fraction of data points, which could be the major possible improvement for the future version of AFS. Though AFS is not applicable for individual-level forgetting due to limitations of algorithm design, we can achieve favorable forgetting outcomes when operating on a batch of query data. In real-world scenarios, companies with millions of users can easily meet the required number of the query dataset. Furthermore, when hospitals or companies face continuous requests for forgetting individual patient data, they can collect and store these requests, and perform the forgetting as a batch once the amount of data reaches the required threshold. Finally, the current version of AFS could

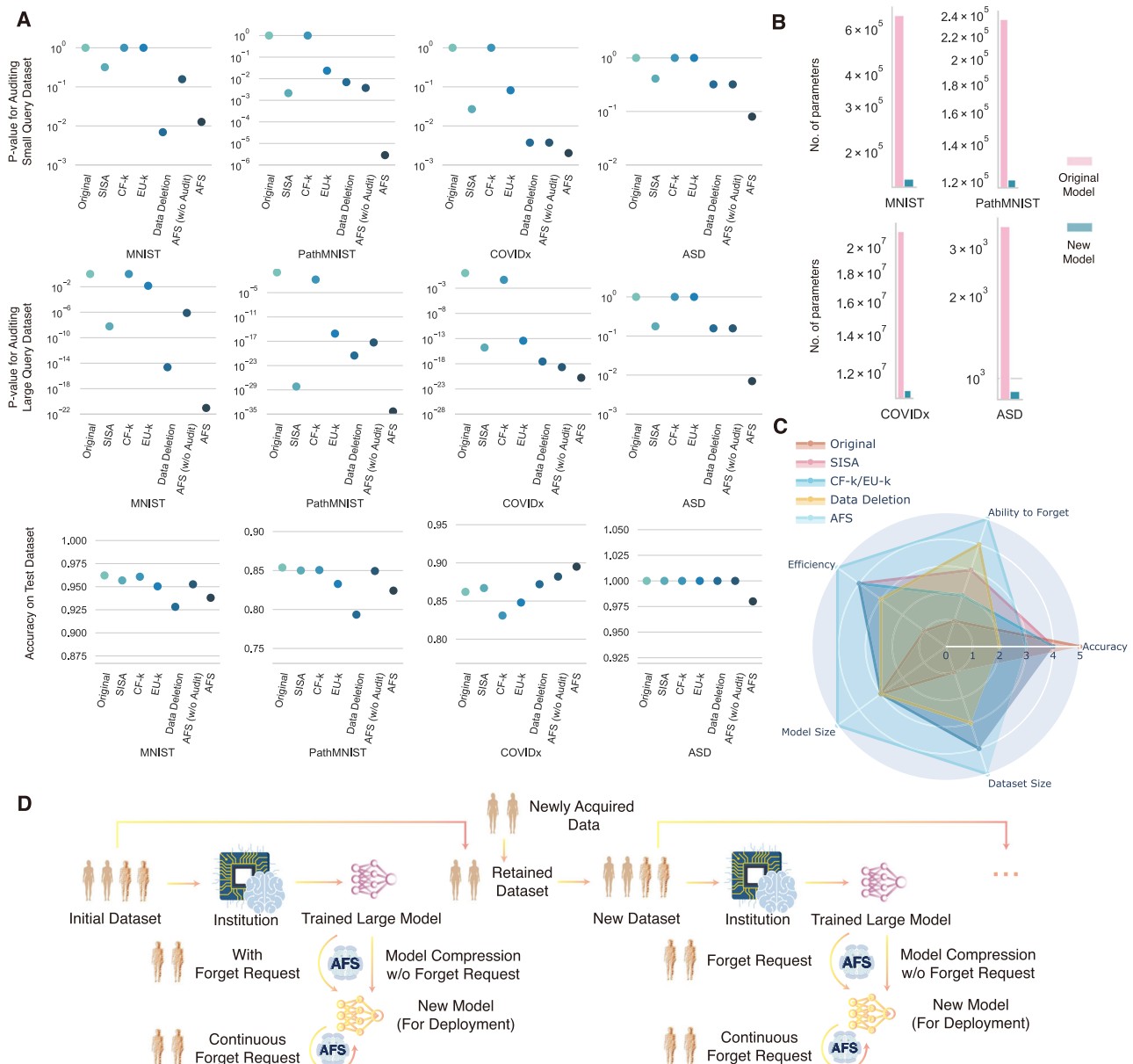

**Fig. 5 | Performance of forgetting using AFS on four datasets. a** The *p* value of auditing on a small query dataset and a large query dataset (QF) and the accuracy of models trained with different methods, including Original (Independent teacher trained with the complete training dataset), SISA, CF-k, EU-k, Data Deletion (the Independent student model trained with partial training dataset and $k = 0.5$), AFS (w/o Audit) and AFS. *p* values were calculated using two-tailed Student's *t* test. **b** The number of parameters for the original large model and the new small model generated by AFS. **c** The qualitative evaluation of three methods, including Original

(Independent teacher trained with the complete training dataset), SISA, CF-k, EU-k, Data Deletion (the Independent student model trained with partial training dataset and $k = 0.5$), and AFS on five dimensions (Ability to forget, accuracy, size of dataset needed for training, size of the generated model and the efficiency of training). A larger value means a stronger ability to forget, higher model accuracy, a smaller size of dataset needed for training, a smaller size of the generated model, and better efficiency of training. **d** Illustration of how to incorporate AFS into real-world applications. The brain icons in **d** are Designed by macrovector/Freepik.

not be applicable for regression tasks because the design of auditing metrics does not work for regression tasks and KP is limited to classification tasks only. Despite these limitations, we believe that AFS will make a valuable contribution toward better protection of people's privacy and the right to revoke data with the rapid development of intelligent healthcare.

## Methods

### The overall framework of AFS

AFS is a unified method to revoke patients' private data by using auditing to guide the forgetting process in a negative feedback manner (Fig. 1).

To audit the membership of the query dataset, AFS takes a pre-trained DL model and the query dataset as inputs, and determines whether the query dataset has been used for training the target DL model. This function was re-implemented based on EMA[41], a published MIA-based method to evaluate the membership of a query dataset. Our re-implementation allows quicker and easier usage of auditing by introducing parallel computing in each epoch, which suggests a significant acceleration for a complete forgetting process and could be more attractive to institutions to forget larger-scale data (Fig. S2).

To forget the query dataset from a DL model, AFS takes the pre-trained DL model and the query dataset to be forgotten as inputs, in which the query dataset has been used to train the DL model. To

effectively forget the information of the query dataset from the pre-trained DL model, an idea is to transfer the information of the remaining dataset except for the query dataset from the pre-trained model to a new model. Therefore, we designed a mechanism called knowledge purification (KP) by using auditing to guide the forgetting process to exclude the information of the query dataset while transferring the remaining information by incorporating the auditing loss into the training process (Fig. 2). With KP integrated, AFS could generate a new model, in which the information of the target dataset should be forgotten under the guidance of auditing.

To provide an applicable solution, we implemented AFS as open-source software that provides a user-friendly entry point allowing users to use both functions with only one command. To demonstrate the generality of AFS, we applied it to four tasks based on four datasets, including the MNIST dataset, the PathMNIST dataset, the COVIDx dataset and the ASD dataset, which have different data sizes (Fig. 3) and various architectures of deep learning networks.

### Dataset preparation

We used four public datasets that were commonly acknowledged in the machine learning and intelligent healthcare field to demonstrate the versatility of AFS. For the benchmark experiment, we applied AFS on MNIST[44] and PathMNIST[45] from the MedMNIST[46] dataset. The MNIST dataset contains 60,000 training images and 10,000 testing images of handwritten digits with size $28 \times 28$ and labeled from 0 to 9. PathMNIST contains 100,000 nonoverlapping image patches from hematoxylin and eosin-stained histological images and 7180 image patches from different clinical centers. In total, 9 types of tissues are involved in the PathMNIST dataset, including adipose, background, debris, lymphocytes, mucus, smooth muscle, normal colon mucosa, cancer-associated stroma, and COAD epithelium. All images in PathMNIST were $224 \times 224$ ($0.5\,\mu m\,px^{-1}$) and were normalized with the Macenko method[47]. For the application of AFS in intelligent healthcare, we used the COVIDx[48] dataset, which contains 13,975 chest X-ray (CXR) images across 13,870 patient cases, and the autism spectrum disorder (ASD) dataset for toddlers[49], which contains 20 features of 1054 samples to be utilized for determining influential autistic traits and improving the classification of ASD cases.

For each dataset, we further sampled partial data as the training dataset, the testing dataset, and the calibration dataset as below:

**MNIST.** We randomly sampled 10,000 images as the training dataset and 10,000 images as the testing dataset. We also randomly sampled 100, 1000, 2000, and 5000 images that are disjoint with the training dataset as four calibration datasets to illustrate the effect of the calibration dataset of varied sizes on auditing and forgetting.

**PathMNIST.** We randomly sampled 10,000 images as the training dataset and 5000 images as the testing dataset. We also randomly sampled 1000 images that are disjoint with the training dataset as the calibration dataset.

**COVIDx.** We randomly sampled 5000 images as the training dataset and 1000 images as the testing dataset. We also randomly sampled 1000 images that are disjoint with the training dataset as the calibration dataset.

**ASD.** We randomly sampled 500 images as the training dataset and 100 images as the testing dataset. We also randomly sampled 100 images that are disjoint with the training dataset as the calibration dataset.

For all four datasets, we randomly sampled partial data from the training dataset with percentage $k$ from {0.25, 0.5, 0.75} as the training dataset for knowledge distillation (KD) and AFS.

In addition, we prepared query datasets with different sizes $N$ from {1, 10, 100, 500, 1000, 2000}. A query dataset that completely overlapped with the training dataset is labeled as QO, while the query dataset that is completely disjoint with the training dataset is labeled QNO. To further understand the effect of the purity of the query dataset, we also prepared the query dataset called QM with a $k$ percentage of the query dataset to be overlapped with the training dataset. Finally, for the query dataset designed to be forgotten, we labeled it as QF. QO, QNO, QM, and QF were all sampled randomly from the complete dataset and all reported values were the average of 5 replicate experiments.

### Deep learning models and experiment setup

To present the generalizability of AFS toward various DL models, we adopted different architectures for each of the four tasks, including the multilayer perception[50] (MLP), the convolutional neural network (CNN)[51] and ResNet[52]. There were a large DL model and a small DL model for each task, where the large model refers to the original pre-trained model and the small model is the new model generated by AFS.

For the MNIST dataset, we used MLP with 671,754 parameters as the teacher model and 155,658 parameters as the student model to achieve the 10-class classification task.

For the PathMNIST dataset, we adopted CNN with 21,285,698 parameters as the teacher model and 11,177,538 parameters as the student network for the 9-class classification task.

For the COVIDx dataset, we took ResNet34 with 21,285,698 parameters as the teacher model and ResNet18 with 11,177,538 parameters as the student network to achieve the binary classification of healthy people and patients.

For the ASD dataset, we used the MLP with 3586 parameters as the teacher model and the MLP with 898 parameters as the student model for the binary classification of autism in toddlers.

During model training, the number of epochs was fixed to 50, the learning rate was set to 1e−5 and the Adam optimizer was used. A workstation with 252 GB RAM, 112 CPU cores and 2 Nvidia V100 GPUs were adopted for all experiments. The AFS method was developed based on Python3.7, PyTorch1.9.1 and CUDA11.4. A detailed list of dependencies could be found in our code availability.

### Audit the membership of query dataset

EMA[41] is designed as a 2-step process. In the first step, the best threshold for each metric is selected to optimize $(TPR(t) + TNR(t))/2$ based on the calibration dataset as shown in Algorithm 1. Once the thresholds for all metrics are selected, the membership of each sample in the query dataset will be confirmed as at least one metric is larger than the corresponding threshold. In total, three metrics, including correctness[53], confidence[54,55], and entropy[56,57], were adopted to further calculate the $p$ value, which is the key audit metric in AFS as proposed in the previous work[41,58]. The details for calculating correctness, confidence, and entropy are as below:

Correctness: the target model is trained to predict correctly on training data and may not generalize well on test data. Thus, we can define the correctness as in Eq. (1):

$$I_{\text{correctness}}(F, (x, y)) = \mathbb{1}\{\arg\max_i F(x)_i = y\} \tag{1}$$

where $F$ is the deep learning model, $\boldsymbol{x}$ is the input data, $F(\boldsymbol{x})$ is the output logits, $y$ is the label, and $\mathbb{1}$ is the indicator function.

Confidence: the target model is usually more confident in predictions on training data, but less confident in test data. Thus, we can define confidence as in Eq. (2):

$$I_{\text{confidence}}(F, (\boldsymbol{x}, y)) = \mathbb{1}\{F(\boldsymbol{x})_y \geq \tau_y\} \tag{2}$$

**Algorithm 1.** Infer thresholds

**Input**: The calibration dataset $D_{cal}$, the pre-trained DL model $A$ and $n$ different metrics $(m_1, \ldots, m_n)$ for membership testing.

**Procedure:**

1 Split $D_{cal}$ into training dataset $D_{cal}^{train}$ and test dataset $D_{cal}^{test}$

2 The calibration model is trained as $f_{D_{cal}} \leftarrow A(D_{cal}^{train})$

3 For $m_i \in \{m_1, \ldots, m_n\}$ do in parallel

4 Compute metrics for training dataset as $M_{train} \leftarrow \{m_i(f_{D_{cal},s}|s \in D_{cal}^{train})\}$

5 Compute metrics for test dataset as $M_{test} \leftarrow \{m_i(f_{D_{cal},s}|s \in D_{cal}^{test})\}$

6 Find $t_i \in argmax_{t \in [M_{train}, M_{test}]}\left(\frac{TPR(t)+TNR(t)}{2}\right)$, where $TPR(t) = \sum_{s \in D_{cal}^{train}} \mathbb{1}\{m_i(s) \geq t\}/|D_{cal}^{train}|$ and $TNR(t) = \sum_{s \in D_{cal}^{test}} \mathbb{1}\{m_i(s) < t\}/|D_{cal}^{test}|$

7 Return The thresholds $t_1, \ldots, t_n$ for $n$ metrics

---

**Algorithm 2.** AFS

**Input**: The calibration dataset $D_{cal}$, the query dataset to forget $D_{forget}$, the sampled training dataset $D_{train}$ for KP, the pre-trained DL model $F$, the new model $f$, and number of epochs $T$.

**Procedure:**

8 For epoch $\in \{1, \ldots, T\}$

9 Forward $D_{train}$ with $F$

10 Forward $D_{train}$ with $f$

11 Infer threshold with $D_{cal}$ on $f$ and audit $D_{forget}$ on $f$ to get $loss_{audit}$

12 Calculate $loss_{AFS} = loss_{classification} + loss_{KD} + loss_{audit}$

13 Update $f$ based on $loss_{AFS}$

14 Return The new student model $f$ with information about $D_{forget}$ forgotten.

---

where $F$ is the deep learning model, $\boldsymbol{x}$ is the input data, $F(\boldsymbol{x})$ is the output logits, $y$ is the label, $F(\boldsymbol{x})_y$ is the output logits for label $y$, $\tau_y$ is a threshold for the logit for label $y$, and $\mathbb{1}$ is the indicator function.

Entropy: the target model is trained by minimizing the prediction loss over training data and usually has a larger prediction entropy on a test sample. Thus, we can define entropy as in Eq. (3):

$$I_{\text{entropy}}(F, (x, y)) = \mathbb{1}\left\{-\sum_i F(x)_i \log(F(x)_i) \leq \hat{\tau}_y\right\} \qquad (3)$$

where $F$ is the deep learning model, $\boldsymbol{x}$ is the input data, $F(\boldsymbol{x})$ is the output logits, $y$ is the label, $F(\boldsymbol{x})_i$ is the output logits for class $i$, $\hat{\tau}_y$ is a threshold, and $\mathbb{1}$ is the indicator function.

Once the membership of all samples in the query dataset is confirmed in the previous step, the query dataset will be further evaluated to determine whether the query dataset has been used to train the target pre-trained DL model. A two-sample statistical test is adopted to evaluate the query dataset based on the sample-wise membership and an all-one vector. The $p$ value of the two-sample statistical test is used as the output of auditing. Given a user-defined threshold $\alpha$, if $p < \alpha$, then users could conclude that the query dataset was not used for training the target DL model. EMA was re-implemented and integrated into AFS to allow easy and fast auditing.

### Audit-guided forgetting of query dataset with AFS

Forgetting aims to remove the remembered information of the query dataset from the target DL model. Similar to knowledge distillation (KD), a teacher-student paradigm was also adopted in AFS, but with an additional requirement to selectively forget information associated with the data we want to forget. Thus, we designed an approach called knowledge purification (KP), meaning purifying the knowledge in the teacher model (the original pre-trained model), discarding the information related to the data that needed to be forgotten and transferring the purified information into the student model (the new model). AFS unified auditing and forgetting into a circular process to effectively enhance the unlearning in a negative feedback manner.

As shown in Fig. 1, during each epoch of training, the training data will be fed into both the teacher model and the student model, while the data to be forgotten will be audited on the student model. Our main goal is to transfer the knowledge from the teacher model to the student model while forcing the student model to reject the information associated with data to be forgotten. In order to achieve that, we added the audit loss into the total loss, thus allowing the student model to accept partial knowledge from the teacher model and achieve KP as shown in Algorithm 2.

### Evaluation metrices

Since all four tasks are either multi-classes classification tasks or binary classification tasks, we adopted the accuracy and F1-score as the evaluation metrics as in Eqs. (4) and (5):

$$\text{Accuracy} = \frac{TP + TN}{TP + TN + FP + FN} \qquad (4)$$

$$F1 - score = \frac{2TP}{2TP + FP + FN} \qquad (5)$$

where TP represents true positives, TN stands for true negatives, FN represents false negatives and FP stands for false positives.

To evaluate the membership of the query dataset, the $p$ value of the two-sample statistical test was used as mentioned previously.

## Reporting summary

Further information on research design is available in the Nature Portfolio Reporting Summary linked to this article.

## Data availability

All four datasets used in this work are publicly available. The MNIST dataset is available at https://www.kaggle.com/datasets/hojjatk/mnist-dataset. The PathMNIST dataset is available at https://medmnist.com/. The COVIDx dataset is stored at https://www.kaggle.com/datasets/andyczhao/covidx-cxr2?select=competition test. The ASD dataset can be accessed at https://www.kaggle.com/datasets/fabdelja/autism-screening-for-toddlers. All data supporting the findings described in this paper are available in the article and in the Supplementary Information and from the corresponding author upon request. Source data are provided with this paper.

## Code availability

The AFS software is publicly available at https://github.com/JoshuaChou2018/AFS and https://doi.org/10.5281/zenodo.8275769. SISA is implemented based on the codes at https://github.com/cleverhans-lab/machine-unlearning.

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

## Acknowledgements
J.Z., H.L., X.L., B.Z., W.H., Z.L., L.Z. and X.G. were supported in part by grants from the Office of Research Administration (ORA) at King Abdullah University of Science and Technology (KAUST) under award number FCC/1/1976-44-01, FCC/1/1976-45-01, REI/1/5202-01-01, REI/1/5234-01-01, REI/1/4940-01-01, RGC/3/4816-01-01, and REI/1/0018-01-01. X.L. was also supported in part by grants from the National Natural Science Foundation of China under grant number No. 62002388.

## Author contributions
Conceptualization: J.Z. and X.G. Design: J.Z. and X.G. Data analysis and interpretation: J.Z., H.L. and W.H. Code implementation: J.Z. and H.L. Application: J.Z., H.L., X.L. and B.Z. Code improvement: Z.L. and L.Z. Drafting of the manuscript: J.Z. and H.L. Critical revision of the manuscript for important intellectual content: J.Z., X.L. and B.Z. Supervision: J.Z. and X.G. Funding acquisition: X.G.

## Competing interests
The authors declare no competing interests.
