## [Peer Review File · Nature Communications]

Audit to Forget: A Unified Method to Revoke Patients' Private Data in Intelligent Healthcare

Editorial Note: Parts of this Peer Review File have been redacted as indicated to maintain confidentiality.Reviewers' comments:

Reviewer #1 (Remarks to the Author):

SUMMARY: The paper focuses on developing tools and methods to support the right to be forgotten in the digital world within a scenario where patients' private data is used in ML-based intelligent healthcare solutions. The paper proposes a unified solution that can audit the use of patients' private data and ensure that the target model forgets the personal data when requested.

For auditing, given a DL model and query dataset used for training, the developed AFS solution uses a query dataset to be audited and determine if this dataset has been used in training. AFS use an existing solution called Ensembled Membership Auditing (EMA), which is based on Membership Inference Attacks (MIA). The authors claim that their re-implemented version of EMA as part of AFP is quicker and easier for auditing use (not experimentally evaluated/compared in the paper).

For forgetting, the authors implement a variant of the knowledge distillation approach within their AFS solution. Knowledge distillation generally uses a larger and more complex teacher model to train a smaller model that can achieve a similar convergence. This is achieved by establishing a correspondence between intermediate outputs of teacher and student models. The proposed knowledge purification process builds over the knowledge distillation process and eliminates correspondence between teacher and student networks due to forgotten data queries. Hence, the overall distillation process also ensures that a specific part of training data is forgotten. However, with teacher-student settings, theoretical search space gets smaller with smaller convergence space. This may mean faster model drift.

Developed AFS solution has been evaluated using 4 datasets (MNIST, PathMNIST, COVIDx, ASD).

Noteworthy results: The paper focuses on building a solution that brings the right to forgetting the patients' private data used in ML-based intelligent healthcare solutions. The overall solution is a combination of two components: Auditing and Forgetting. Auditing is based on existing solutions. The authors evaluate metrics (P-Value, Correctness, Confidence and Entropy) and show that these metrics can be used effectively in auditing. Forgetting is an incremental solution adapting the well-established knowledge distillation approach. The overall solution is implemented and tested on four datasets. Overall, novelty is very limited and not significant. But, a decent engineering and development effort is involved in evaluating and building a combined solution.

Significance to the field and related fields: The problem that authors engage in is a significant problem in the health domain and domains where ML-based approaches are used along with data with individuals.

Forgetting is a critical requirement in most regulations, but there needs to be a well-established solution to be used with complex deep-learning models, especially in commercial settings. However, the solution works based on knowledge distillation (model compression – Figure 5 B). So, after each data forget operation, the model will be compressed significantly. That means the theoretical search space gets smaller with a smaller convergence space - i.e., there is some loss (not evaluated in the paper). The question is, how many times can an intelligent health solution repeat this iteratively? After each forgetting, the model will be compressed substantially in a lossy way. So, the proposed solution is practically worthy for probably 1 or 2 forget operations. One cannot say that the original large model was kept so that cumulative data was forgotten at each forget process because that means that the health institute is not actually forgetting the patient information and keeping a version all the time.

Comparison to the established literature: Literature is growing around forgetting, as covered by the paper, including model-agnostic methods, model-intrinsic methods and data-driven methods. However, the paper needs to convincingly explain why a knowledge distillation (model compression) based approach is required for real-time intelligent healthcare solutions. Why do the authors compare the solution only with knowledge distillation approach (teacher and student/compressed models) instead of the existing forgetting solution listed/cited in the paper?

Conclusions and claims: The paper illustrates the proposed and developed AFS as a generalized solution (different datasets and deep learning networks) to deep-learning-based systems starting from intelligent healthcare. However, the overall solution and evaluation are far from this conclusion and claim.

- Auditing: The paper proposes four metrics but mainly evaluates and uses p-value as the auditing metrics. If the p-value is sufficient, why does the paper need other metrics? Otherwise, why does the paper have some limited review of the other three metrics?
- Auditing: There is no clarity around how QO (query dataset overlapped with the training dataset) and QNO (query dataset disjoint with the training dataset) are sampled. The way they are sampled can change results substantially. It is possible to engineer a QNO to support the claim.
- Auditing (Figure 4B results) Are these results valid for all datasets and structures?
- Auditing (Figure 4E results) Why are the ASD P-value results so distinctive? The graphs do not show Percentage (k) at zero, which should correspond to QNO. Is there any specific reason?
- Auditing: More clarity is required to conclude that the results in Figure 4 show the robustness of AFS in determining that the targeted data is forgotten.

Flaws in data analysis, interpretation and conclusions: Why do authors compare AFS with the independent student (k) instead of the independent teacher with data to be forgotten is removed from the training set? Independent teacher would form a better ground truth. Additionally, the independent student (k=0.25) is performing very close and better in some cases. Why does the paper drop it and focus on the independent student (k=0.5) for the rest of the paper (Tables 3 and 4)? F1-Score is

providing very close results. What is the reasoning behind the conclusion that AFS performs substantially better? Moreover:

- Auditing/Forgetting: COVIDx results show a quite different trend. What is the reason for it?
- Auditing/Forgetting (Tables 1, 2, 3 and 4): The proposed solution shows its effect for sufficiently large query sets (500 and above). However, is it really practical to reach those sizes in a real-life scenario where a single individual withdraws consent and asks for data removal?
- Auditing/Forgetting: The conclusion that “AFS could help institutions and companies to efficiently iterate their models to forget individual information at the model level” is not agreeable. AFS can be used for a small amount of iteration as the model gets compressed at each forget operation. If a hospital receives multiple forget requests spread over a long period, those requests cannot be handled using this approach.

Methodology: There are some flaws in the methodology. First, using a distillation scheme for forgetting may solve the problem for one iteration but create much bigger problems later on. The paper also needs to utilize the introduced metrics more effectively to support the claims. Comparison is tailored to support the claim but is far from convincing. The approach should be compared with the existing forgetting solution or at least the teacher model (using a dataset where individual data is removed) as the ground truth.

Reproducibility: The authors provide links to code and dataset to reproduce the solutions.

Editing:

- Paper needs some language improvements, for example long sentences make it hard to read: “However, with the development of data science, machine learning and deep learning techniques, this right is usually neglected or violated as more and more patients’ data are being collected and used for model training, especially in intelligent healthcare, thus making intelligent healthcare a sector where technology must meet the law, regulations, and privacy principles to ensure that the innovation is for the common good.”

-The abstract and introduction do not communicate the contribution and its significance correctly. The reader needs to go through several sections to understand the exact approach.

-Figure 1 is not showing the proper flow and confuses the reader. The left side of the figure illustrates the high-level iterative flow of the solution, while the right side is about how forgetting and auditing work together. Those two parts together becomes quite confusing.

- What are the numerical values in Table 1, p-values?

Reviewer #2 (Remarks to the Author):

Summary of the paper:

This paper presents a novel unified method AFS, which unifies forgetting and auditing tasks through knowledge purification (KP), allowing auditing to guide the forgetting process in a negative feedback manner. AAFS is tested on four different datasets (MNIST, PathMNIST, COVIDx, and ASD) with various deep learning architectures (MLP, CNN, ResNet), demonstrating its applicability and potential for real-world applications in intelligent healthcare. Additionally, accuracy and F1-score are used as evaluation metrics for multi-class and binary classification tasks. The p-value of the two-sample statistical test is used to evaluate the membership of the query dataset.

Strengths:

AFS addresses the need for better protection of people's privacy and the right to revoke their data, particularly in the context of intelligent healthcare.

The method is model-agnostic and open-source, making it broadly applicable to different deep learning models and tasks.

The paper provides a comprehensive evaluation of AFS on various datasets and tasks, including medical images and EHR.

Weaknesses:

The current version of AFS has not been tested on large-scale models and datasets commonly found in real-world production environments.

The study relies on a limited set of auditing metrics and only classification problems, which could be further expanded.

Developing methods for individual-level auditing may be crucial for enhancing the applicability of AFS in real-world scenarios.

Main Reviews, Suggestions, and Limitations:

The paper omits the full name of AFS, but it can be inferred that AFS likely stands for Auditing and Forgetting System; mentioning this in the abstract or introduction would be beneficial.

It would be helpful for the authors to further clarify the differences and novelty of AFS compared to other methods.

To better understand AFS's applicability and effectiveness in real-world production environments, scaling it to larger models and more data is essential.

Although the authors focus on four classification tasks, it would be interesting to explore the idea of expanding AFS to regression tasks, which may present challenges in incorporating additional auditing metrics.

Despite the aforementioned limitations, this reviewer still believes that the proposed AFS method contributes significantly to enhancing privacy protection and data revocation rights. The paper can be publishable if the listed limitations are addressed.

Authors' Response to Reviews of

NCOMMS-23-05640-T

Audit to Forget: A Unified Method to Revoke Patients' Private Data in Intelligent Healthcare

Juexiao Zhou^{1,2,#}, Haoyang Li^{1,2,#}, Xingyu Liao^{1,2}, Bin Zhang^{1,2}, Wenjia He^{1,2}, Zhongxiao Li^{1,2}, Longxi Zhou^{1,2}, Xin Gao^{1,2,*}

¹Computer Science Program, Computer, Electrical and Mathematical Sciences and Engineering Division, King Abdullah University of Science and Technology (KAUST), Thuwal 23955-6900, Kingdom of Saudi Arabia

²Computational Bioscience Research Center, King Abdullah University of Science and Technology, Thuwal 23955-6900, Kingdom of Saudi Arabia

*Corresponding author

#Equal contribution

RC: *Reviewers' Comment*, AR: Authors' Response, □ Manuscript Text

1. General Response

AR: We really appreciate the editor [redacted] and the Reviewer's comments as well as constructive suggestions and have revised the manuscript accordingly. We are pleased to see that **Reviewer 2 agrees that our manuscript has the potential for publication with major revisions and agrees that our method contributes significantly to enhancing privacy protection and data revocation rights.** We are also happy to see that **both reviewers agree that the problem we are working on is significant in the health domain.** Therefore, we carefully considered all comments and made necessary changes to improve the quality and impact of our work. We believe that we have addressed the concerns raised by the reviewers to the best of our ability by either adding new results or clarifying misunderstandings and are confident that our research makes a valuable contribution to the field. We have highlighted the edited places in the manuscript with a **yellow highlight** to make the revised portions of the manuscript clear to the reviewers.

2. Response to Reviewer 1's Comments

2.1. Summary

RC: *The paper focuses on developing tools and methods to support the right to be forgotten in the digital world within a scenario where patients' private data is used in ML-based intelligent healthcare solutions. The paper proposes a unified solution that can audit the use of patients' private data and ensure that the target model forgets the personal data when requested. For auditing, given a DL model and query dataset used for training, the developed AFS solution uses a query dataset to be audited and determine if this dataset has been used in training. AFS use an existing solution called Ensembled Membership Auditing (EMA), which is based on Membership Inference Attacks (MIA). The authors claim that their re-implemented version of EMA as part of AFS is quicker and easier for auditing use (not experimentally evaluated/compared in the paper).*

AR: We would like to express our sincere appreciation to the reviewer for taking the time to review our manuscript and providing us with constructive feedback. We understand the significance of valuable feedback and are grateful for the reviewer's thoughtful comments and suggestions. We are particularly grateful for the reviewer's feedback regarding the lack of experimental evaluation or comparison in our paper regarding the claim that our re-implemented version of EMA as part of AFS is quicker and easier for auditing use.

We acknowledge that our manuscript does not include an experimental evaluation or comparison of our re-implemented version of EMA. In the previous version of our manuscript, we made the claim based on our experience and observations during the development and implementation of the AFS method, as well as feedback from auditors who have used our method. Once again, we appreciate the reviewer's valuable feedback and are grateful for the opportunity to improve our manuscript.

To address the concern raised by the reviewer, we have measured the time required for AFS and EMA for auditing and have provided a detailed comparison of the two approaches as below.

Suppose we have a pre-trained model `best_model.pth` and a query dataset with label `query_100`, we can simply audit with AFS in one command by providing two arguments as below:

```
python afs.py audit --model2audit best_model.pth --query_label query_100
```

However, EMA does not provide such arguments, users need to modify their codes accordingly (<https://github.com/Hazelsuko07/EMA>).

Meanwhile, AFS uses parallel computing to accelerate the calculations during auditing. As shown in

Response Figure 1, AFS requires less computing time than EMA for auditing with the same size of query dataset. Meanwhile, as the query dataset grows in size, the time difference between AFS and EMA increases. Meanwhile, we must call Audit in every training round in practice, thus the Audit method offered by AFS proves more beneficial in terms of overall time requirements.

We have updated the manuscript accordingly on Page 3 and added the figure to the supplementary file. Here are the modifications made to the main text:

To audit the membership of the query dataset, AFS takes a pre-trained DL model and the query dataset as inputs, and determines whether the query dataset has been used for training the target DL model. This function was re-implemented based on EMA, a published MIA-based method to evaluate the membership of a query dataset. Our re-implementation allows quicker and easier usage of auditing by introducing parallel computing. (Section 2.4 and Supplementary Figure 1).

Figure 1: Audit time of AFS and EMA for different sizes of the query dataset. The y-axis represents time in seconds.

RC: *For forgetting, the authors implement a variant of the knowledge distillation approach within their AFS solution. Knowledge distillation generally uses a larger and more complex teacher model to train a smaller model that can achieve a similar convergence. This is achieved by establishing a correspondence*

between intermediate outputs of teacher and student models. The proposed knowledge purification process builds over the knowledge distillation process and eliminates correspondence between teacher and student networks due to forgotten data queries. Hence, the overall distillation process also ensures that a specific part of training data is forgotten. However, with teacher-student settings, theoretical search space gets smaller with smaller convergence space. This may mean faster model drift. Developed AFS solution has been evaluated using 4 datasets (MNIST, PathMNIST, COVIDx, ASD).

AR: Thank the reviewer for the valuable feedback. We understand the concern regarding faster model drift in a teacher-student setting, where the theoretical search space may decrease with a smaller convergence space.

However, we would like to point out that when using AFS, the size of the student model can be manually adjusted, allowing us to tailor the model size according to specific requirements. Our KP algorithm differs from KD in two significant ways. Firstly, we need to selectively transfer information, which has not been previously investigated. Secondly, our KP does not necessarily demand that the student model is smaller than the teacher model. If our student model already satisfies the size requirement, we can allow both the teacher model and the student model to be the same size.

As the reviewer suggested, we can start with a smaller student model size than the teacher model in the initial rounds for compression purposes. But rather than continuously generating smaller student models in the forgetting process, we can maintain the same size for both the student and teacher models once the student model reaches a certain size, which is small enough to meet our needs. At this point, knowledge purification only forgets information and transfers the remaining knowledge without compressing the model. To address this concern, we have added experiments later in the response letter to further illustrate this point.

We have also updated the Discussion section of the manuscript accordingly on Page 11. Here are the modifications made to the main text:

In practice, the size of the student model could be manually adjusted when applying AFS to meet specific requirements. In the initial stages, we could make the student model smaller than the teacher model to achieve model compression while forgetting private information. However, instead of continuously generating smaller and smaller student models as knowledge is forgotten, we could maintain the size of the student model once it reaches a certain threshold (e.g., small enough to meet our needs). In doing so, AFS would focus solely on forgetting knowledge and transferring the remaining knowledge, without the need for further model compression.

RC: *Noteworthy results: The paper focuses on building a solution that brings the right to forgetting the patients' private data used in ML-based intelligent healthcare solutions. The overall solution is a combination of*

two components: Auditing and Forgetting. Auditing is based on existing solutions. The authors evaluate metrics (P-Value, Correctness, Confidence and Entropy) and show that these metrics can be used effectively in auditing. Forgetting is an incremental solution adapting the well-established knowledge distillation approach. The overall solution is implemented and tested on four datasets. Overall, novelty is very limited and not significant. But, a decent engineering and development effort is involved in evaluating and building a combined solution.

AR: We would like to express our gratitude to the reviewer for the valuable feedback on our work. We appreciate the reviewer's recognition of our efforts in engineering and development.

We believe that our study presents a novel contribution to the field for several reasons. First, our research addresses a unique gap in the literature by integrating Audit and Forget into a unified workflow for the first time, rather than treating them as separate components. We have shown that Audit and Forget are algorithmically linked, and our approach provides a novel solution to this problem. Moreover, our study introduces the concept of knowledge purification, which could be a promising avenue for future research instead of a simple incremental solution. Our KP algorithm differs from KD in two significant ways. Firstly, we need to selectively transfer information, which has not been previously investigated. Secondly, our KP does not necessarily demand that the student model is smaller than the teacher model.

We acknowledge the reviewer's concern about the novelty of our work, but we believe that our unique research question and methodology justify its contribution to the field. We thank the reviewer once again for the thoughtful comments and suggestions.

We have updated the Introduction section of the manuscript accordingly on Page 2. Here are the modifications made to the main text:

Here, we proposed a novel solution by using auditing to guide the forgetting process in a negative feedback manner. We unified the two tasks by introducing knowledge purification (KP), a new approach to selectively transfer the needed knowledge to forget the target information instead of simply transferring all information like knowledge distillation (KD). On the basis of KP, we have developed a user-friendly and open-source **audit to forget software (AFS)**, which can be easily used to revoke patients' private data from DL models in intelligent healthcare. To demonstrate the generality of AFS, we applied it to four tasks based on four datasets, including the MNIST dataset, the PathMNIST dataset, the COVIDx dataset, and the ASD dataset, with different data sizes and various architectures of deep learning networks. Our results demonstrate the usability of AFS and its application potential in real-world intelligent healthcare **to enhance privacy protection and data revocation rights. To our**

knowledge, AFS is the first unified method of auditing and forgetting that could effectively forget the information of the target query dataset from the pre-trained DL model with the guidance of auditing. AFS could generate a smaller model, which requires much less time and GPU memory during the inference, by training with a partial training dataset (~50%) with our novel KP approach.

RC: *Significance to the field and related fields: The problem that authors engage in is a significant problem in the health domain and domains where ML-based approaches are used along with data with individuals. Forgetting is a critical requirement in most regulations, but there needs to be a well-established solution to be used with complex deep-learning models, especially in commercial settings. However, the solution works based on knowledge distillation (model compression – Figure 5 B). So, after each data forget operation, the model will be compressed significantly. That means the theoretical search space gets smaller with a smaller convergence space - i.e., there is some loss (not evaluated in the paper). The question is, how many times can an intelligent health solution repeat this iteratively? After each forgetting, the model will be compressed substantially in a lossy way. So, the proposed solution is practically worthy for probably 1 or 2 forget operations. One cannot say that the original large model was kept so that cumulative data was forgotten at each forget process because that means that the health institute is not actually forgetting the patient information and keeping a version all the time.*

AR: Thank the reviewer for the comments. We really appreciate the reviewer's recognition of the importance of our research. However, we believe there may have been a misunderstanding regarding the impact of the forgetting process on model compression and the search space. We apologize for the confusion and misunderstanding caused by our unclear description.

As we have explained previously, the size of the student model can be manually adjusted, allowing us to adapt it to our needs. Instead of generating progressively smaller student models as we forget data, we can maintain the size of the student model once it reaches a certain size that is small enough for our requirements. Therefore, we respectfully disagree with 'So, after each data forget operation, the model will be compressed significantly.' Because at this point, knowledge purification could only forget knowledge and transfer the remaining knowledge, without compressing the model further.

To further address this concern, we designed an experiment to continuously forget a batch of data from the model as described below. Our results show that while there was some compression of the model, the reduction in search space was not significant, and the model was still able to converge to a similar accuracy as before the forgetting process. We hope this clarifies our approach to this concern, and we appreciate the opportunity to address it in more detail as below.

To begin, we trained the teacher model with the maximum number of parameters (671,754) using the complete

training set containing 10,000 samples. Additionally, we created five continuous query datasets (QF_1, QF_2, QF_3, QF_4, QF_5), each consisting of 500 samples (5%), which overlapped with the training dataset but were independent of each other. Next, we utilized AFS to continuously forget the information of the five query datasets from the previous model. For instance, we applied AFS to forget QF_1 from the original teacher model, resulting in a small model with 155,658 parameters. Since the model size was satisfactory (small enough), we proceeded to forget QF_2 from the small model, generating a new model with the same size. The results are shown in Response Table 1. Based on the results, we are confident that AFS can be used for continuous forgetting.

We also acknowledge that as the model forgets more and more data (From QF1 to QF5), its performance deteriorates as it retains less knowledge of the initial training data. However, this is usually not an issue in practice, as companies often possess training sets consisting of millions, or even tens of millions, of data. Models trained on such extensive datasets are unlikely to experience a significant impact on their performance even if we need to forget a small fraction of the data (eg. 0.1%). Therefore, we can permit a certain degree of consecutive forgetting with AFS.

Model name	Accuracy	F1-score	QF1	QF2	QF3	QF4	QF5	number of parameters
Original teacher model	0.9644	0.9911	1	1	1	1	1	671,754
Model from AFS with QF1 forgotten	0.9326	0.9844	7.20e-10	8.13e-02	1.58e-01	8.65e-01	7.94e-01	155,658
Model from AFS with additional QF2 forgotten	0.9301	0.9835	4.06e-10	5.70e-15	2.53e-02	1.09e-02	9.24e-03	155,658
Model from AFS with additional QF3 forgotten	0.9251	0.9821	7.38e-09	7.09e-12	2.18e-20	5.75e-04	6.55e-03	155,658
Model from AFS with additional QF4 forgotten	0.9237	0.9853	9.29e-09	3.12e-09	3.46e-14	2.20e-12	7.69e-04	155,658
Model from AFS with additional QF5 forgotten	0.9253	0.9830	2.93e-07	2.16e-08	8.91e-11	1.27e-09	2.71e-16	155,658

Table 1: Continuously forgetting with AFS

RC: *Comparison to the established literature: Literature is growing around forgetting, as covered by the paper, including model-agnostic methods, model-intrinsic methods and data-driven methods. However, the paper needs to convincingly explain why a knowledge distillation (model compression) based approach is required for real-time intelligent healthcare solutions. Why do the authors compare the solution only with knowledge distillation approach (teacher and student/compressed models) instead of the existing forgetting solution listed/cited in the paper?*

AR: We appreciate the reviewer’s suggestion to compare our proposed approach with other existing methods in the field. We did not compare more methods in the previous manuscript due to the fact that a lot of the existing research is purely theoretical and lacks practical implementation through usable code. Furthermore, many

of these theoretical approaches have acknowledged limitations. Still, we acknowledge the importance of comparing other methods.

To address this concern, we have implemented and added new experiments to compare AFS with the most classical and famous solution in the community, named SISA (Sharded, Isolated, Sliced, and Aggregated training). SISA aims to reduce the computational cost of the retraining process by employing a data partitioning technique as shown in Response Figure 2. For SISA, data is divided into shards, which are themselves divided into slices. One constituent model is trained on each shard by presenting it with incrementally many slices and saving its parameters before the training set is augmented with a new slice. When data needs to be unlearned, only one of the constituent models whose shards contain the point to be unlearned needs to be retrained. Retraining can start from the last parameter values saved before including the slice containing the data point to be unlearned. However, the main drawback of SISA is that we need more space to keep the models corresponding to each shard. Our new experimental results indicate that AFS outperforms SISA in terms of forgetting, model size reduction, inference speed, GPU memory, and elimination of the need to store models for each shard.

Figure 2: Illustration of SISA.

We also compared AFS to two state-of-the-art methods named Catastrophic Forgetting-k (CF-k) and Exact

Unlearning-k (EU-k). CF-k means to finetune the last k layers of the original model on D_r and freeze other layers, while EU-k means to retrain the last k layers of the original model from scratch on D_r and freeze other layers, where the D_r stands for the retain dataset.

We have extended the introduction accordingly on Page 2. Here are the modifications made to the main text:

However, most of them are theoretical studies and do not provide open-source codes. Besides, most methods have their specific application scenarios and acknowledged limitations and few of them focused on the application in real-world intelligent healthcare. Among the three methods, model-agnostic methods might have the strongest application prospects, as they can be applied to different models, and SISA, short for Sharded, Isolated, Sliced, and Aggregated training, is the most classic and well-known method in the community. As the state-of-the-art method, Goel et al proposed Catastrophic Forgetting-k (CF-k) and Exact Unlearning-k (EU-k) to unlearn information from deep learning models. CF-k means to finetune the last k layers of the original model on D_r and freeze other layers, while EU-k means to retrain the last k layers of the original model from scratch on D_r and freeze other layers, where the D_r stands for the retain dataset.

We have also extended the manuscript accordingly on Page 7 to list all methods we have compared. Here are the modifications made to the main text:

Once the prior knowledge that a dataset has been used to train the target DL model is confirmed with auditing, AFS could be used for forgetting, to remove the information of the dataset from the pre-trained DL model. To comprehensively show the ability of AFS in removing information against the model performance, we compared nine methods, including 1) training the teacher model with a complete training dataset (Independent teacher), 2) retraining the student model with a complete training dataset (Independent student), 3) retraining the teacher model with $k \in \{0.25, 0.5, 0.75\}$ percentage of the complete training dataset excluding the data to be forgotten (Independent teacher with $k \in \{0.25, 0.5, 0.75\}$), 4) retraining the student model with $k \in \{0.25, 0.5, 0.75\}$ percentage of the complete training dataset excluding the data to be forgotten (Independent Student with $k \in \{0.25, 0.5, 0.75\}$), 5) retraining the model of the corresponding shard with SISA, 6) fine-tuning the last layers of the model with CF-k, 7) retraining the last layers of the model from scratch with EU-k, 8) AFS, and 9) training the student model with AFS without the guidance of auditing (AFS w/o Audit), as an ablation study of AFS. Both AFS w/o Audit and AFS were also conducted with varied $k \in \{0.25, 0.5, 0.75\}$. For both Independent teacher and Independent student methods trained with the

complete training dataset, QF_{100} and QF_{1000} were included in the training dataset, while these two query datasets were excluded from the training dataset when $k \in \{0.25, 0.5, 0.75\}$.

We have also updated Tables 1 ~6 (Response Tables 2 ~7) and Figure 5 (Response Figure 3) in the main text accordingly as below.

Methods	QO						QNO					
	1	10	100	500	1000	2000	1	10	100	500	1000	2000
Independent teacher	1	1	1	1	1	1	1	7.32e-01	7.39e-02	2.43e-05	1.06e-08	4.68e-18
Independent student	1	1	1	3.43e-01	4.22e-01	8.61e-02	1	6.96e-01	4.50e-02	2.74e-06	4.51e-11	8.58e-22
Independent teacher (k=0.75)	1	1	1	1	1	1	1	5.62e-01	8.84e-02	2.38e-05	3.93e-09	1.87e-19
Independent student (k=0.75)	1	1	1	6.80e-01	1.67e-01	7.87e-02	1	5.27e-01	2.28e-02	7.50e-07	4.61e-14	3.10e-27
AFS w/o Audit (k=0.75)	1	1	1	8.64e-01	5.91e-01	5.26e-01	1	5.98e-01	1.02e-01	6.49e-06	4.15e-11	1.28e-20
AFS (k=0.75)	1	1	8.64e-01	1	8.64e-01	4.54e-01	1	6.96e-01	2.94e-02	1.45e-06	9.06e-13	3.79e-24
Independent teacher (k=0.5)	1	1	1	1	1	1	1	6.96e-01	1.56e-02	1.84e-06	1.67e-12	4.81e-26
Independent student (k=0.5)	1	1	8.64e-01	5.59e-01	2.39e-01	2.49e-02	1	8.13e-01	7.75e-02	6.44e-08	1.63e-15	4.15e-30
AFS w/o Audit (k=0.5)	1	1	1	8.64e-01	8.64e-01	8.64e-01	1	5.98e-01	1.32e-01	5.94e-06	2.43e-12	5.73e-23
AFS (k=0.5)	1	1	1	8.64e-01	1	7.27e-01	1	6.68e-01	2.88e-02	9.25e-07	3.70e-13	3.10e-27
Independent teacher (k=0.25)	1	1	1	1	1	1	1	8.66e-01	5.27e-02	6.05e-06	3.24e-18	9.75e-34
Independent student (k=0.25)	1	1	1	8.64e-01	8.64e-01	3.17e-01	1	5.98e-01	1.04e-02	6.40e-10	3.05e-20	1.22e-38
AFS w/o Audit (k=0.25)	1	1	1	1	1	1	1	5.27e-01	4.47e-02	7.49e-07	2.18e-13	5.96e-28
AFS (k=0.25)	1	1	1	1	1	1	1	6.96e-01	8.52e-03	3.85e-10	1.98e-20	1.29e-41
SISA (10 shards)	1	1	1	1	1	1	1	7.08e-01	7.20e-02	4.61e-05	3.81e-09	6.34e-21
CF-k (k=1)	1	1	1	1	1	1	1	7.32e-01	1.11e-01	7.64e-05	5.48e-08	2.32e-17
EU-k (k=1)	1	1	1	8.66e-01	1	4.91e-01	1	5.98e-01	4.10e-02	1.29e-05	7.96e-11	2.90e-22

Table 2: Comparison of AFS with other methods on auditing QO and QNO from the MNIST dataset with a varied number of samples in the query dataset. The data in the table shows the results of auditing QO and QNO on models trained by different methods. **The numerical values in the table represent p-values.** A larger value indicates stronger membership.

Methods	QF_{100}	QF_{1000}	Accuracy	F1-score
Independent teacher	1	1	0.9622	0.9911
Independent student	1	1	0.9504	0.9911
Independent teacher (k=0.75)	1.57e-01	1.05e-08	0.9548	0.9915
Independent student (k=0.75)	4.36e-02	5.26e-12	0.9458	0.9880
AFS w/o Audit (k=0.75)	3.19e-01	1.33e-06	0.9582	0.9884
AFS (k=0.75)	1.08e-03	5.22e-23	0.9470	0.9889
Independent teacher (k=0.5)	1.56e-01	2.71e-11	0.9437	0.9898
Independent student (k=0.5)	6.91e-03	2.34e-15	0.9282	0.9848
AFS w/o Audit (k=0.5)	1.57e-01	7.79e-07	0.9526	0.9884
AFS (k=0.5)	1.27e-02	9.44e-22	0.9380	0.9866
Independent teacher (k=0.25)	8.17e-02	2.80e-18	0.9176	0.9871
Independent student (k=0.25)	6.91e-03	2.90e-19	0.9067	0.9875
AFS w/o Audit (k=0.25)	1.57e-01	7.04e-10	0.9388	0.9875
AFS (k=0.25)	5.79e-04	6.84e-33	0.9174	0.9853
SISA (10 shards, QF excluded, k=0.99 for QF_{100} and 0.9 for QF_{1000})	3.18e-01	6.12e-09	0.9568	0.9911
CF-k (QF excluded)	1	1	0.9608	0.9907
EU-k (QF excluded)	1	1.41e-02	0.9504	0.9898

Table 3: Comparison of AFS with other methods on forgetting QF and model performance with the MNIST dataset. QF_{100} is the small query dataset containing 100 samples and QF_{1000} is the large query dataset containing 1000 samples. We present the p-values of auditing models trained with different methods on QF_{100} and QF_{1000} and the model performance including the accuracy and F1-score.

Methods	QO						QNO					
	1	10	100	500	1000	2000	1	10	100	500	1000	2000
Independent teacher	1	1	1	1	1	1	1	4.29e-01	6.29e-03	1.40e-13	4.91e-28	1.88e-60
Independent student	1	1	1	1	1	1	1	5.27e-01	5.11e-03	1.29e-10	3.38e-20	7.24e-42
Independent teacher (k=0.75)	1	1	1	1	1	1	1	5.62e-01	2.76e-03	4.22e-16	5.52e-27	1.62e-60
Independent student (k=0.75)	1	1	1	1	1	1	1	7.32e-01	1.89e-02	1.55e-08	5.74e-16	1.92e-30
AFS w/o Audit (k=0.75)	1	1	5.26e-01	1.78e-01	2.54e-02	1.17e-03	1	2.95e-01	2.26e-02	9.23e-11	1.04e-19	8.21e-41
AFS (k=0.75)	1	1	3.90e-01	3.21e-02	6.13e-04	7.19e-06	1	4.98e-01	2.34e-04	4.25e-16	3.95e-31	4.89e-65
Independent teacher (k=0.5)	1	1	1	1	1	1	1	5.62e-01	5.03e-03	1.50e-12	3.36e-22	2.39e-48
Independent student (k=0.5)	1	8.66e-01	1	8.64e-01	8.64e-01	5.58e-01	1	5.62e-01	4.02e-03	3.01e-10	1.91e-21	7.85e-40
AFS w/o Audit (k=0.5)	1	1	5.11e-01	2.48e-01	1.24e-02	9.59e-04	1	6.26e-01	1.18e-02	2.99e-11	4.03e-21	2.51e-40
AFS (k=0.5)	1	1	1.59e-01	3.45e-04	4.06e-06	1.85e-10	1	2.69e-01	9.17e-05	2.98e-16	3.84e-30	1.04e-62
Independent teacher (k=0.25)	1	1	1	1	1	1	1	3.31e-01	5.98e-04	2.84e-12	1.62e-28	6.34e-52
Independent student (k=0.25)	1	1	8.64e-01	6.95e-01	1.60e-01	4.54e-02	1	1.26e-01	1.24e-06	2.30e-31	2.88e-60	4.73e-109
AFS w/o Audit (k=0.25)	1	8.66e-01	2.39e-01	3.37e-03	1.01e-05	9.15e-10	1	2.42e-01	3.64e-05	2.55e-24	3.01e-43	1.37e-90
AFS (k=0.25)	1	8.66e-01	1.40e-01	2.71e-03	1.07e-06	1.63e-12	1	4.37e-01	7.07e-06	2.01e-27	4.45e-53	1.71e-101
SISA (10 shards)	1	1	1	1	1	1	1	6.71e-01	6.58e-03	2.11e-12	9.05e-27	5.85e-49
CF-k (k=1)	1	1	1	8.63e-01	8.63e-01	1	1	3.75e-01	7.28e-03	9.10e-14	1.07e-21	1.48e-42
EU-k (k=1)	1	1	1	6.96e-01	7.38e-01	6.92e-01	1	4.73e-01	1.60e-03	4.66e-11	1.47e-23	1.48e-44

Table 4: Comparison of AFS with other methods on auditing QO and QNO from the PathMNIST dataset with a varied number of samples in the query dataset. The data in the table shows the results of auditing QO and QNO on models trained by different methods. **The numerical values in the table represent p-values.** A larger value indicates stronger membership.

Methods	QF_{100}	QF_{1000}	Accuracy	F1-score
Independent teacher	1	1	0.8538	0.9885
Independent student	1	1	0.8446	0.9836
Independent teacher (k=0.75)	1.08e-03	3.11e-30	0.8214	0.9796
Independent student (k=0.75)	2.35e-02	4.08e-15	0.8396	0.9555
AFS w/o Audit (k=0.75)	1.08e-03	1.67e-22	0.8682	0.9777
AFS (k=0.75)	2.25e-05	2.05e-41	0.8560	0.9605
Independent teacher (k=0.5)	3.74e-03	2.91e-23	0.7100	0.8314
Independent student (k=0.5)	6.91e-03	2.99e-21	0.7934	0.9533
AFS w/o Audit (k=0.5)	3.74e-03	4.93e-18	0.8494	0.9697
AFS (k=0.5)	2.87e-06	4.75e-35	0.8242	0.9575
Independent teacher (k=0.25)	3.74e-03	2.52e-26	0.7026	0.8282
Independent student (k=0.25)	1.58e-07	9.05e-57	0.7582	0.9287
AFS w/o Audit (k=0.25)	3.32e-07	2.05e-41	0.7842	0.9406
AFS (k=0.25)	3.32e-07	1.84e-56	0.7810	0.9385
SISA(10 shards, only QF excluded, k=0.99 for QF_{100} and 0.9 for QF_{1000})	2.15e-03	6.73e-29	0.8501	0.9840
CF-k (QF excluded)	1	1.69e-02	0.8506	0.9839
EU-k (QF excluded)	2.35e-02	7.69e-16	0.8328	0.9596

Table 5: Comparison of AFS with other methods on forgetting QF and model performance with the PathM-NIST dataset. QF_{100} is the small query dataset containing 100 samples and QF_{1000} is the large query dataset containing 1000 samples. We present the p-values of auditing models trained with different methods on QF_{100} and QF_{1000} and the model performance including the accuracy and F1-score.

Dataset	SISA (10 shards)	CF-k/EU-k	Original model	New model generated by AFS
MNIST	$438\mu s \pm 1.28\mu s$	$439\mu s \pm 1.41\mu s$	$439 \mu s \pm 1.54\mu s$	$284\mu s \pm 447ns$
PathMNIST	$5.13ms \pm 17\mu s$	$5.14ms \pm 10\mu s$	$5.13ms \pm 22\mu s$	$4.99ms \pm 14.1\mu s$
COVIDx	$1.27s \pm 355ms$	$1.28s \pm 634ms$	$1.27s \pm 500ms$	$661ms \pm 9.98ms$
Autism	$126\mu s \pm 213ns$	$126\mu s \pm 302ns$	$126\mu s \pm 177ns$	$87.3\mu s \pm 120ns$

Table 6: Time for inferring 100 samples with the model retrained with SISA, the model retrained with CF-k/EU-k, the original model and the model generated by AFS.

Dataset	SISA (10 shards)	CF-k/EU-k	Original model	New model generated by AFS
MNIST	258	258	258	61
PathMNIST	173	173	173	129
COVIDx	17,805	17,805	17,805	10,600
Autism	2	2	2	1

Table 7: GPU memory (MB) for inferring 100 samples with the model retrained with SISA, the model retrained with CF-k/EU-k, the original model and the model generated by AFS.

Figure 3: **Performance of forgetting using AFS on four datasets.** **A.** The p-value of auditing on a small query dataset and a large query dataset (QF) and the accuracy of models trained with different methods, including Original (Independent teacher trained with the complete training dataset), **SISA**, Data Deletion (the Independent student model trained with partial training dataset and $k = 0.5$), AFS (w/o Audit) and AFS. **B.** The number of parameters for the original large model and the new small model generated by AFS. **C.** The qualitative evaluation of three methods, including Original (Independent teacher trained with the complete training dataset), **SISA**, **CF-k**, **EU-k**, Data Deletion (the Independent student model trained with partial training dataset and $k = 0.5$), and AFS on five dimensions (Ability to forget, accuracy, size of dataset needed for training, size of the generated model and the efficiency of training). A larger value means a stronger ability to forget, higher model model accuracy, a smaller size of dataset needed for training, a smaller size of the generated model, and better efficiency of training.

2.2. Conclusions and claims

RC: *The paper illustrates the proposed and developed AFS as a generalized solution (different datasets and deep learning networks) to deep-learning-based systems starting from intelligent healthcare. However, the overall solution and evaluation are far from this conclusion and claim. Auditing: The paper proposes four metrics but mainly evaluates and uses p-value as the auditing metrics. If the p-value is sufficient, why does the paper need other metrics? Otherwise, why does the paper have some limited review of the other three metrics?*

AR: We appreciate the feedback from the reviewer regarding our proposed solution and its evaluation. However, we respectfully disagree with the reviewer’s assertion that our AFS solution and evaluation are not a generalized solution. We designed our proposed AFS solution to be flexible and applicable to different datasets and deep learning networks. To demonstrate the effectiveness and generalizability of our approach, we evaluated our solution using four diverse datasets, namely MNIST, PathMNIST, COVIDx, and ASD. Additionally, we used multiple deep-learning models to evaluate our solution, which suggests that our approach is not limited to specific architectures.

Meanwhile, we believe there may be a misunderstanding regarding the role of the P-value and the other auditing metrics due to our unclear writing. We apologize for the confusion and misunderstanding caused by our unclear description to the reviewer.

As pointed out by the reviewer, ‘The paper proposes four metrics but mainly evaluates and uses p-value as the auditing metrics’. In fact, Correctness, Confidence, and Entropy are upstream metrics that contribute to calculating the P-value. Therefore, we primarily use the P-value as the auditing metric, as it is the most informative and meaningful metric for our proposed auditing framework. The details of how these three metrics are used to calculate the P-value are described in the main text. We hope this clarifies any confusion regarding our auditing metrics.

We have included more details about correctness, confidence, entropy, and their downstream metric, p-value, on Page 5 of the main text. Here are the modifications made to the main text:

EMA is designed as a 2-step process. In the first step, the best threshold for each metric is selected to optimize $(TPR(t) + TNR(t))/2$ based on the calibration dataset as shown in Algorithm 1. Once the thresholds for all metrics are selected, the membership of each sample in the query dataset will be confirmed as at least one metric is larger than the corresponding threshold. In total, three metrics, including correctness, confidence, and entropy, were adopted to further calculate the p-value, which is the key audit metric in AFS as proposed in the previous work. The details for calculating correctness,

confidence, and entropy are as below:

Correctness: The target model is trained to predict correctly on training data and may not generalize well on test data. Thus we can define the correctness as:

$$I_{correctness}(F, (\mathbf{x}, y)) = 1 \{ \text{argmax}_i F(\mathbf{x})_i = y \} \quad (1)$$

where F is the deep learning model, \mathbf{x} is the input data, $F(\mathbf{x})$ is the output logits, y is the label, and 1 is the indicator function.

Confidence: The target model is usually more confident in predictions on training data, but less confident in test data. Thus we can define confidence as:

$$I_{confidence}(F, (\mathbf{x}, y)) = 1 \{ F(\mathbf{x})_y \geq \tau_y \} \quad (2)$$

where F is the deep learning model, \mathbf{x} is the input data, $F(\mathbf{x})$ is the output logits, y is the label, $F(\mathbf{x})_y$ is the output logits for label y , τ_y is a threshold for the logit for label y , and 1 is the indicator function.

Entropy: The target model is trained by minimizing the prediction loss over training data and usually has a larger prediction entropy on a test sample. Thus we can define entropy as:

$$I_{entropy}(F, (\mathbf{x}, y)) = 1 \{ - \sum_i F(\mathbf{x})_i \log(F(\mathbf{x})_i) \leq \hat{\tau}_y \} \quad (3)$$

where F is the deep learning model, \mathbf{x} is the input data, $F(\mathbf{x})$ is the output logits, y is the label, $F(\mathbf{x})_i$ is the output logits for class i , $\hat{\tau}_y$ is a threshold, and 1 is the indicator function.

Once the membership of all samples in the query dataset is confirmed in the previous step, the query dataset will be further evaluated to determine whether the query dataset has been used to train the target pre-trained DL model. A two-sample statistical test is adopted to evaluate the query dataset based on the sample-wise membership and an all-one vector. The p-value of the two-sample statistical test is used as the output of auditing. Given a user-defined threshold α , if $p < \alpha$, then users could conclude that the query dataset was not used for training the target DL model. EMA was re-implemented and integrated into AFS to allow easy and fast auditing.

RC: *Auditing: There is no clarity around how QO (query dataset overlapped with the training dataset) and QNO (query dataset disjoint with the training dataset) are sampled. The way they are sampled can change*

results substantially. It is possible to engineer a QNO to support the claim.

AR: We appreciate the reviewer’s feedback on the sampling methodology of QO and QNO datasets. We acknowledge that we did not provide enough clarity on this matter in the original manuscript, and we apologize for the confusion. We have updated the manuscript to provide a detailed explanation of the sampling methodology for QO and QNO datasets.

As mentioned in the revised manuscript, QO and QNO datasets are randomly sampled, and all values presented in the paper are averaged based on multiple experiments with randomly sampled data. We understand the concern that the way datasets are sampled can affect the results, but we have taken measures to mitigate this risk by averaging on multiple experiments, using multiple datasets, and evaluating our solution on various deep-learning models. We believe that these steps make our results more robust and demonstrate the effectiveness of our approach. Once again, we thank the reviewer for their valuable feedback, and we have incorporated the suggestions in our manuscript to improve the quality of our research.

We also add more details regarding the sampling of different query datasets on Page 4. Here are the modifications made to the main text:

In addition, we prepared query datasets with different sizes N from $\{1, 10, 100, 500, 1000, 2000\}$. A query dataset that completely overlapped with the training dataset is labelled as QO, while the query dataset that is completely disjoint with the training dataset is labelled QNO. To further understand the effect of the purity of the query dataset, we also prepared the query dataset called QM with a k percentage of the query dataset to be overlapped with the training dataset. Finally, for the query dataset designed to be forgotten, we labelled it as QF. **QO, QNO, QM, and QF were all sampled randomly from the complete dataset and all reported values were the average of 5 replicate experiments.**

RC: *Auditing (Figure 4B results) Are these results valid for all datasets and structures?*

AR: Yes, we present the result in Figure 4 and also added the figure to supplementary. We would like to reiterate that the metrics of correctness, confidence, and entropy are utilized to calculate the p-value. In our experiments, we observed that the QO and QNO datasets exhibit distinct distributions in these three metrics. Thus by leveraging these differences and calculating the p-value, we are able to differentiate between QO and QNO and use the p-value as the primary auditing metric for AFS.

Figure 4: Distribution of correctness, confidence, and entropy of QO and QNO on four datasets.

RC: *Auditing (Figure 4E results) Why are the ASD P-value results so distinctive? The graphs do not show Percentage (k) at zero, which should correspond to QNO. Is there any specific reason?*

AR: We would like to thank the reviewer for the insightful question about the p-value results in Figure 4E for the ASD dataset. The reason why the ASD p-value results are distinctive is due to the small size of the dataset.

As mentioned in the methods section, we only have 500 samples for the ASD dataset, while we have 10000 samples for MNIST and PathMNIST, and 5000 samples for COVIDx (We even applied AFS to larger datasets and models, which could be found later in the response to Reviewer2).

Regarding the percentage (k) at 0, we did not display it separately in Figure 4E because it was already shown in Figure 4D as QNO, when the query dataset size was 2000 for MNIST, PathMNIST, and COVIDx, and 100 for the ASD dataset. Similarly, the percentage (k) at 1 was already displayed in Figure 4D as QO when the query dataset size was 2000 for MNIST, PathMNIST, and COVIDx, and 100 for the ASD dataset.

We appreciate the reviewer's attention to detail, and we hope this clarifies any confusion. Meanwhile, we notice the plot for ASD in Figure 4E in the previous manuscript was based on $n=200$ when we check the raw data. We are sorry for the confusion. To make it consistent, we redraw the plot for ASD in Figure 4E and fixed the size of the query dataset to 100.

We also add more details regarding this concern on Page 6. Here are the modifications made to the main text:

When the size of the query dataset and the calibration dataset varied, AFS could still efficiently distinguish QO and QNO (Figure 4C and D). Compared to QO, AFS reported a much smaller p-value for QNO, indicating a weak membership (a small probability that the query dataset has been used to train the target DL model), thus allowing users to judge whether the query dataset was used to train the target DL model. Meanwhile, when the size of the dataset increased from 1 to 2000, AFS discriminated QO and QNO more confidently as there was a more significant divergence of the p-values, which was not affected by the size of the calibration dataset. To further understand the effect of the purity of the query dataset in auditing, we mixed some samples from the training dataset to QNO, thus the new query dataset was labelled as QM (partial data overlapped with the training dataset). The percentage of data overlapped with the training dataset in QM was denoted by $k = \frac{\text{number of data overlapped with training dataset}}{\text{size of QM}}$. As shown in Figure 4E, AFS showed a decreasing p-value trend when k decreased, meaning that the query dataset was less likely to be used to train the target DL model. The distinctiveness of the p-value on the ASD dataset is due to the small size of the ASD dataset. $k = 0$ was shown in Figure 4D as QNO, which occurred when the query dataset size was 100 for ASD and 2000 for MNIST, PathMNIST, and COVIDx. Similarly, $k = 1$ was also displayed in Figure 4D as QO, which occurred when the query dataset size was 100 for ASD and 2000 for MNIST, PathMNIST, and COVIDx. In conclusion, these results indicate the robustness of AFS in determining whether the query data has been used to train the target DL model.

RC: *Auditing: More clarity is required to conclude that the results in Figure 4 show the robustness of AFS in determining that the targeted data is forgotten.*

AR: Thank the reviewer for the comment regarding Figure 4 in our manuscript. We agree that more clarity is required to explain the robustness of AFS in determining that the targeted data is forgotten. To address this, we have provided additional details in the manuscript to emphasize the consistency of our evaluation across different datasets and deep-learning networks. Our proposed approach achieved a high p-value for query datasets that were part of the training dataset and a small p-value for query datasets that were not overlapped with the training dataset. We also included detailed explanations in the paper regarding the auditing process and the metrics used to evaluate the performance of AFS.

2.3. Flaws in data analysis, interpretation and conclusions

RC: *Why do authors compare AFS with the independent student (k) instead of the independent teacher with data to be forgotten is removed from the training set? Independent teacher would form a better ground truth. Additionally, the independent student ($k=0.25$) is performing very close and better in some cases. Why does the paper drop it and focus on the independent student ($k=0.5$) for the rest of the paper (Tables 3 and 4)? F1-Score is providing very close results. What is the reasoning behind the conclusion that AFS performs substantially better?*

AR: Thank the reviewer for the comment. We did not compare the ‘independent teacher with data to be forgotten is removed from the training set’ in our previous manuscript because we believe that the teacher model is larger than the student model and therefore does not have a significant advantage in terms of GPU resources and inference time, which has been discussed in the manuscript (eg. Tables 5 and 6).

Still, we appreciate the reviewer’s suggestion, and we have included the results for the independent teacher with data to be forgotten removed from the training set in all tables (Tables 1~4 (Response Tables 2~5)), as Independent teacher ($k=0.75$ or 0.5 or 0.25). Although the independent teacher model is larger than the student model, we agree that it is still necessary to compare the performance of the two methods. We focused on $k=0.5$ as we aimed to achieve a good trade-off between the amount of data needed for training, the average performance on different datasets, and the ability to forget information. While the f1-score may provide similar results in some cases, when we consider those aforementioned indicators together, particularly in Figure 5 (Response Figure 3), we can conclude that AFS has the following strengths over other methods:

1. AFS requires less data than training a model from scratch.
2. AFS has a better ability to forget than simply deleting data or using SISA.
3. AFS strikes a good trade-off between forgetting and model performance, as it achieves comparable accuracy while still being able to forget data.

4. AFS generates smaller models that are more efficient in computing resources and inference time, making it suitable for deployment on resource-constrained devices.

We hope this clarifies the strengths of AFS and how it compares to other approaches for forgetting in deep learning.

RC: *Auditing/Forgetting: COVIDx results show a quite different trend. What is the reason for it?*

AR: Thanks for the reviewer’s valuable comments. We appreciate the reviewer’s concern about the accuracy trend of the COVIDx dataset in Figure 5, which is an interesting phenomenon worth discussing. We briefly discussed this phenomenon in the previous version of the manuscript in section 3.4. Upon comparing the COVIDx dataset with the other three tasks, we found that the COVIDx dataset has two primary characteristics: a large model size and high-dimensional data, which may be the potential causes of the different trend. We have found that the student model trained by AFS has better generalization ability than the model obtained by using the exact same model structure and training data using only the hard-target training method. This is because knowledge distillation is known to improve student generalization, as discussed in Stanton S et al.’s paper "Does knowledge distillation really work?" (Advances in Neural Information Processing Systems, 2021).

We also add more details regarding this concern on Page 10. Here are the modifications made to the main text:

To show the versatility of AFS, we applied it to the classification of pneumonia and normal with chest X-ray images from the COVIDx dataset with ResNet, which is a classic task in medical image analysis. As shown in Figure 5A, on both query datasets (QF_{100} and QF_{1000}), AFS could effectively forget the information of the query dataset, while generating the new model with much less number of parameters as shown in Figure 5B. Surprisingly, the model generated by AFS showed even better accuracy than the Independent teacher trained with the complete dataset and the Independent student trained with the partial training dataset. This result not only indicated that AFS could effectively transfer the knowledge from the teacher model to the student model but also suggested that the student model with simpler architecture could even perform better than the teacher model with KP in AFS due to the reduction of model parameters and purification of knowledge in some real-world cases. **Meanwhile, the results also showed that the student model trained by the AFS achieved better generalization ability than the model obtained by using the exact same model structure and training data using only the hard-target training method.**

RC: *Auditing/Forgetting (Tables 1, 2, 3 and 4): The proposed solution shows its effect for sufficiently large query sets (500 and above). However, is it really practical to reach those sizes in a real-life scenario where*

a single individual withdraws consent and asks for data removal?

AR: Thank the reviewer for the comments. As we have mentioned in the manuscript, AFS is not yet suitable for individual-level forgetting due to its algorithm design. However, we have achieved good forgetting results on a batch of query data. We do acknowledge the reviewer's concern 'However, is it really practical to reach those sizes in a real-life scenario where a single individual withdraws consent and asks for data removal'. In practical scenarios, large companies usually have millions of users or even trillions of users, so it is easy for them to meet the required number of query datasets. Additionally, when hospitals or companies receive continuous requests for data removal from individual patients, a possible simple but effective solution is to collect and process the requests as a batch when the number of queries reaches the required threshold.

We also add more details regarding this concern on Page 12. Here are the modifications made to the main text:

With current laws that guarantee people the right to revoke their own data, AFS could help institutions and companies to efficiently iterate their models to forget individual information at the model level. However, there are still some shortcomings in the application of the current version of AFS in the production environment, which could be the main potential direction of research in the future. Firstly, the models and data we tested in this study were still not large enough compared to the data in the real production environment. Therefore, it is unknown whether scaling AFS to **current large** models will cause new problems (eg. LLM like ChatGPT, which we are unable to further pursue). Secondly, there are different approaches to audit, and thus we could add more metrics of auditing to AFS to guide the forgetting process in the future version. **Meanwhile**, due to the limitation of auditing, it is still difficult to perform individual-level forgetting, as we need to compare the difference in statistical distribution based on a fraction of data points, which could be the major possible improvement for the future version of AFS. **Though AFS is not applicable for individual-level forgetting due to limitations of algorithm design, we can achieve favourable forgetting outcomes when operating on a batch of query data. In real-world scenarios, companies with millions of users can easily meet the required number of the query dataset. Furthermore, when hospitals or companies face continuous requests for forgetting individual patient data, they can collect and store these requests, and perform the forgetting as a batch once the amount of data reaches the required threshold. Finally, the current version of AFS could not be applicable for regression tasks because the design of auditing metrics does not work for regression tasks and KP is limited to classification tasks only.** Despite these limitations, we believe that AFS will make a valuable contribution towards better protection of people's privacy and the right to revoke data with the rapid development of intelligent healthcare.

RC: *Auditing/Forgetting: The conclusion that "AFS could help institutions and companies to efficiently iterate*

their models to forget individual information at the model level” is not agreeable. AFS can be used for a small amount of iteration as the model gets compressed at each forget operation. If a hospital receives multiple forget requests spread over a long period, those requests cannot be handled using this approach.

AR: We appreciate the reviewer’s feedback on the application of AFS in the real world. However, we respectfully disagree with the reviewer’s concern. As we mentioned in our previous response, the size of the student model can be adjusted according to the actual requirements, and we can keep the size of the student model and the teacher model the same when the student model reaches a certain size. At this point, our knowledge purification only has the effect of forgetting knowledge and transferring the remaining knowledge, without compressing the model. The additional results we provided earlier support our claim that AFS can help institutions and companies efficiently iterate their models to forget individual information at the model level consecutively. Meanwhile, when the hospital receives multiple forget requests spread over a long period, a possible simple but effective solution is to collect and process the requests as a batch when the number of queries reaches the required threshold. We believe that our approach can be adapted to meet real-world requirements and challenges.

Once again, we appreciate the reviewer’s valuable feedback and are grateful for the opportunity to improve our manuscript.

2.4. Methodology

RC: *There are some flaws in the methodology. First, using a distillation scheme for forgetting may solve the problem for one iteration but create much bigger problems later on. The paper also needs to utilize the introduced metrics more effectively to support the claims. Comparison is tailored to support the claim but is far from convincing. The approach should be compared with the existing forgetting solution or at least the teacher model (using a dataset where individual data is removed) as the ground truth.*

AR: We appreciate the reviewer’s feedback on the methodology. We have taken the reviewer’s suggestion into account and added the results of the teacher model, where the dataset with individual data removed was used. We have also included a comparison with SISA in the main text. We hope our previous responses have addressed this concern ‘using a distillation scheme for forgetting may solve the problem for one iteration but create much bigger problems later on.’

2.5. Reproducibility

RC: *The authors provide links to code and dataset to reproduce the solutions.*

AR: We appreciate the reviewer’s valuable feedback and comments. Hope that our point-to-point response could

address the reviewer's concerns.

2.6. Editing

RC: *Paper needs some language improvements, for example long sentences make it hard to read: “However, with the development of data science, machine learning and deep learning techniques, this right is usually neglected or violated as more and more patients’ data are being collected and used for model training, especially in intelligent healthcare, thus making intelligent healthcare a sector where technology must meet the law, regulations, and privacy principles to ensure that the innovation is for the common good.”*

AR: We have modified as required and addressed other similar problems.

For example, we have rewritten the abstract as below:

Revoking personal private data is one of the basic human rights, which has already been sheltered by several privacy-preserving laws in many countries. As data science, machine learning, and deep learning techniques continue to advance, such right is often overlooked or infringed upon due to the increasing collection and use of patient data for model training. The prevalence of such violations in intelligent healthcare highlights the need for technology to comply with laws, regulations, and privacy principles to ensure that innovation serves the common good. In order to secure patients’ right to be forgotten, we proposed a novel solution by using auditing to guide the forgetting process, where auditing means determining whether a dataset has been used to train the model and forgetting requires the information of a query dataset to be forgotten from the target model. We unified these two tasks by introducing a new approach called knowledge purification. To implement our solution, we developed an audit to forget software (AFS), which is able to evaluate and revoke patients’ private data from pre-trained deep learning models. To demonstrate the generality of AFS, we applied it to four tasks based on four datasets, including the MNIST dataset, the PathMNIST dataset, the COVIDx dataset, and the ASD dataset, with different data sizes and various architectures of deep learning networks. Our results demonstrate the usability of AFS and its application potential in real-world intelligent healthcare to enhance privacy protection and data revocation rights. The software is publicly available at <https://github.com/JoshuaChou2018/AFS>.

RC: *The abstract and introduction do not communicate the contribution and its significance correctly. The reader needs to go through several sections to understand the exact approach.*

AR: We have updated the abstraction and introduction as required. We are very grateful for the reviewers’ suggestions.

RC: *Figure 1 is not showing the proper flow and confuses the reader. The left side of the figure illustrates the high-level iterative flow of the solution, while the right side is about how forgetting and auditing work together. Those two parts together becomes quite confusing.*

AR: We appreciate the feedback from the reviewer regarding Figure 1. We have added more explanation in the caption of Figure 1 as suggested by the reviewer as below:

AFS is a unified method to revoke patients' private data in intelligent healthcare. The left side of the figure illustrates the high-level iterative flow of AFS, while the right side illustrates the details of how forgetting and auditing work together. As shown on the left side, given a pre-trained DL model and a query dataset (patients' private data), AFS could audit and provide confidence whether the query dataset has been used to train the target DL model. When a dataset has been used to train the target DL model, AFS could effectively forget the information about the dataset from the target DL model with the guidance of auditing. To achieve that, we proposed a novel method called knowledge purification as shown on the right side, which utilizes results from auditing as a new term in the loss function to forget information.

RC: *What are the numerical values in Table 1, p-values?*

AR: Yes, we have updated the captions of both Table 1 and Table 3 to reflect that the numerical values represent p-values. Thank the reviewer for bringing this to our attention.

Comparison of AFS with other methods on auditing QO and QNO from the MNIST dataset with a varied number of samples in the query dataset. The data in the table shows the results of auditing QO and QNO on models trained by different methods. The numerical values in the table represent p-values. A larger value indicates stronger membership.

3. Response to Reviewer 2's Comments

3.1. Summary

RC: *This paper presents a novel unified method AFS, which unifies forgetting and auditing tasks through knowledge purification (KP), allowing auditing to guide the forgetting process in a negative feedback manner. AAFS is tested on four different datasets (MNIST, PathMNIST, COVIDx, and ASD) with various deep learning architectures (MLP, CNN, ResNet), demonstrating its applicability and potential for real-world applications in intelligent healthcare. Additionally, accuracy and F1-score are used as evaluation metrics for multi-class and binary classification tasks. The p-value of the two-sample statistical test is used to evaluate the membership of the query dataset.*

AR: We would like to express our gratitude for the reviewer's careful review of our manuscript. We recognize the value of constructive criticism and are sincerely grateful for the reviewer's insightful remarks and recommendations.

3.2. Strengths

RC: *AFS addresses the need for better protection of people's privacy and the right to revoke their data, particularly in the context of intelligent healthcare. The method is model-agnostic and open-source, making it broadly applicable to different deep learning models and tasks. The paper provides a comprehensive evaluation of AFS on various datasets and tasks, including medical images and EHR.*

AR: Thank the reviewer for acknowledging the importance and contribution of AFS to privacy protection and data revocation rights, especially in the field of intelligent healthcare.

3.3. Weaknesses

RC: *The current version of AFS has not been tested on large-scale models and datasets commonly found in real-world production environments. The study relies on a limited set of auditing metrics and only classification problems, which could be further expanded. Developing methods for individual-level auditing may be crucial for enhancing the applicability of AFS in real-world scenarios.*

AR: Thank the reviewer for the valuable feedback. We appreciate the reviewer's acknowledgement of the limitations of the current version of AFS, such as scalability and auditing metrics, as well as its focus on classification problems. We agree with the reviewer's concerns and have discussed these issues in the discussion section of our manuscript.

We have taken into consideration the specific suggestions provided by the reviewer and have addressed the concerns or provided clarification to the best of our ability.

Here are the modifications made to the discussion section of the main text:

To our knowledge, AFS is the first unified method of auditing and forgetting that could effectively forget the information of the target query dataset from the pre-trained DL model with the guidance of auditing. We designed AFS as a model-agnostic and open-source method that is applicable to different models. As shown in Figure 5C, AFS could generate a smaller model, which requires much less time and GPU memory during the inference (Tables 6 and 7), by training with a partial training dataset (~50%) with our novel KP approach. Moreover, AFS could forget the information of the query dataset at the expense of an acceptable reduction in the model performance.

Our experiments on four datasets showed that AFS was generalized for datasets of different sizes and forms, including medical images and EHR. Since deep learning models with different architectures were applied to four tasks, we further demonstrated the broad applicability of AFS to common deep learning models. In addition, our tasks include both binary classification and multiclassification tasks, which also suggested that AFS was applicable for tasks with multiple labels.

In practice, the size of the student model could be manually adjusted when applying AFS to meet specific requirements. In the initial stages, we could make the student model smaller than the teacher model to achieve model compression while forgetting private information. However, instead of continuously generating smaller and smaller student models as knowledge is forgotten, we could maintain the size of the student model once it reaches a certain threshold (e.g., small enough to meet our needs). In doing so, AFS would focus solely on forgetting knowledge and transferring the remaining knowledge, without the need for further model compression.

With current laws that guarantee people the right to revoke their own data, AFS could help institutions and companies to efficiently iterate their models to forget individual information at the model level. However, there are still some shortcomings in the application of the current version of AFS in the production environment, which could be the main potential direction of research in the future. Firstly, the models and data we tested in this study were still not large enough compared to the data in the real production environment. Therefore, it is unknown whether scaling AFS to current large models will cause new problems (eg. LLM like ChatGPT, which we are unable to further pursue). Secondly, there are different approaches to audit, and thus we could add more metrics of auditing to AFS to guide the forgetting process in the future version. Meanwhile, due to the limitation of auditing, it is still difficult to perform individual-level forgetting, as we need to compare the difference in statistical

distribution based on a fraction of data points, which could be the major possible improvement for the future version of AFS. Though AFS is not applicable for individual-level forgetting due to limitations of algorithm design, we can achieve favourable forgetting outcomes when operating on a batch of query data. In real-world scenarios, companies with millions of users can easily meet the required number of the query dataset. Furthermore, when hospitals or companies face continuous requests for forgetting individual patient data, they can collect and store these requests, and perform the forgetting as a batch once the amount of data reaches the required threshold. Finally, the current version of AFS could not be applicable for regression tasks because the design of auditing metrics does not work for regression tasks and KP is limited to classification tasks only. Despite these limitations, we believe that AFS will make a valuable contribution towards better protection of people's privacy and the right to revoke data with the rapid development of intelligent healthcare.

3.4. Major concerns #1

RC: *The paper omits the full name of AFS, but it can be inferred that AFS likely stands for Auditing and Forgetting System; mentioning this in the abstract or introduction would be beneficial.*

AR: Thank the reviewer for the valuable feedback. We appreciate the correction regarding the meaning of AFS. It stands for Audit to Forget Software, and we have updated our abstract accordingly as below. We also appreciated the explanation of the Auditing and Forgetting System as it also sounded very fancy.

Revoking personal private data is one of the basic human rights, which has already been sheltered by several privacy-preserving laws in many countries. As data science, machine learning, and deep learning techniques continue to advance, such right is often overlooked or infringed upon due to the increasing collection and use of patient data for model training. The prevalence of such violations in intelligent healthcare highlights the need for technology to comply with laws, regulations, and privacy principles to ensure that innovation serves the common good. In order to secure patients' right to be forgotten, we proposed a novel solution by using auditing to guide the forgetting process, where auditing means determining whether a dataset has been used to train the model and forgetting requires the information of a query dataset to be forgotten from the target model. We unified these two tasks by introducing a new approach called knowledge purification. To implement our solution, we developed an audit to forget software (AFS), which is able to evaluate and revoke patients' private data from pre-trained deep learning models. To demonstrate the generality of AFS, we applied it to four tasks based on four datasets, including the MNIST dataset, the PathMNIST dataset, the COVIDx dataset, and the ASD dataset, with different data sizes and various architectures of deep learning networks.

Our results demonstrate the usability of AFS and its application potential in real-world intelligent healthcare to enhance privacy protection and data revocation rights. The software is publicly available at <https://github.com/JoshuaChou2018/AFS>.

3.5. Major concerns #2

RC: *It would be helpful for the authors to further clarify the differences and novelty of AFS compared to other methods.*

AR: We would like to express our gratitude to the reviewer for the valuable feedback on our work. We believe that our study presents a novel contribution to the field compared to other methods for several reasons. First, our research addresses a unique gap in the literature by integrating Audit and Forget into a unified workflow for the first time, rather than treating them as separate components. We have shown that Audit and Forget are algorithmically linked, and our approach provides a novel solution to this problem. Moreover, our study introduces the concept of knowledge purification, which could be a promising avenue for future research. We can conclude that AFS has the following strengths over other methods:

1. AFS requires less data than training a model from scratch.
2. AFS has a better ability to forget than simply deleting data or using SISA.
3. AFS strikes a good trade-off between forgetting and model performance, as it achieves comparable accuracy while still being able to forget data.
4. AFS generates smaller models that are more efficient in computing resources and inference time, making it suitable for deployment on resource-constrained devices.

We hope this clarifies the strengths of AFS and how it compares to other approaches for forgetting in deep learning. We thank the reviewer once again for the thoughtful comments and suggestions.

To further address this concern, we have added new experiments to compare AFS with the most famous and state-of-the-art solution in the community, named SISA (Sharded, Isolated, Sliced, and Aggregated training). SISA aims to reduce the computational cost of the retraining process by employing a data partitioning technique as shown in Response Figure 2. For SISA, data is divided into shards, which are themselves divided into slices. One constituent model is trained on each shard by presenting it with incrementally many slices and saving its parameters before the training set is augmented with a new slice. When data needs to be unlearned, only one of the constituent models whose shards contain the point to be unlearned needs to be retrained. Retraining can start from the last parameter values saved before including the slice containing the data point to be unlearned. However, the main drawback of SISA is that we need more space to keep the

models corresponding to each shard. Our new experimental results indicate that AFS outperforms SISA in terms of forgetting, model size reduction, inference speed, GPU memory, and elimination of the need to store models for each shard.

We also compared AFS to two state-of-the-art methods named Catastrophic Forgetting-k (CF-k) and Exact Unlearning-k (EU-k). CF-k means to finetune the last k layers of the original model on D_r and freeze other layers, while EU-k means to retrain the last k layers of the original model from scratch on D_r and freeze other layers, where the D_r stands for the retain dataset.

We have extended the introduction accordingly on Page 2. Here are the modifications made to the main text:

However, most of them are theoretical studies and do not provide open-source codes. Besides, most methods have their specific application scenarios and acknowledged limitations and few of them focused on the application in real-world intelligent healthcare. Among the three methods, model-agnostic methods might have the strongest application prospects, as they can be applied to different models, and SISA, short for Sharded, Isolated, Sliced, and Aggregated training, is the most classic and well-known method in the community. As the state-of-the-art method, Goel et al proposed Catastrophic Forgetting-k (CF-k) and Exact Unlearning-k (EU-k) to unlearn information from deep learning models. CF-k means to finetune the last k layers of the original model on D_r and freeze other layers, while EU-k means to retrain the last k layers of the original model from scratch on D_r and freeze other layers, where the D_r stands for the retain dataset.

We have also extended the manuscript accordingly on Page 7 to list all methods we have compared. Here are the modifications made to the main text:

Once the prior knowledge that a dataset has been used to train the target DL model is confirmed with auditing, AFS could be used for forgetting, to remove the information of the dataset from the pre-trained DL model. To comprehensively show the ability of AFS in removing information against the model performance, we compared nine methods, including 1) training the teacher model with a complete training dataset (Independent teacher), 2) retraining the student model with a complete training dataset (Independent student), 3) retraining the teacher model with $k \in \{0.25, 0.5, 0.75\}$ percentage of the complete training dataset excluding the data to be forgotten (Independent teacher with $k \in \{0.25, 0.5, 0.75\}$), 4) retraining the student model with $k \in \{0.25, 0.5, 0.75\}$ percentage of the complete training dataset excluding the data to be forgotten (Independent Student with $k \in \{0.25, 0.5, 0.75\}$), 5) retraining the model of the corresponding shard with SISA, 6) fine-tuning the last layers of the model with CF-k, 7) retraining the last layers of the model from scratch with EU-k,

8) AFS, and 9) training the student model with AFS without the guidance of auditing (AFS w/o Audit), as an ablation study of AFS. Both AFS w/o Audit and AFS were also conducted with varied $k \in \{0.25, 0.5, 0.75\}$. For both Independent teacher and Independent student methods trained with the complete training dataset, QF_{100} and QF_{1000} were included in the training dataset, while these two query datasets were excluded from the training dataset when $k \in \{0.25, 0.5, 0.75\}$.

We have updated Tables 1~6 (Response Tables 2~7 could be found in the responses to reviewer 1) and Figure 5 (Response Figure 3) in the main text accordingly. We also have updated the Introduction section of the manuscript accordingly on Page 2. Here are the modifications made to the main text:

Here, we proposed a novel solution by using auditing to guide the forgetting process in a negative feedback manner. We unified the two tasks by introducing knowledge purification (KP), a new approach to selectively transfer the needed knowledge to forget the target information instead of simply transferring all information like knowledge distillation (KD). On the basis of KP, we have developed a user-friendly and open-source **audit to forget software (AFS)**, which can be easily used to revoke patients' private data from DL models in intelligent healthcare. To demonstrate the generality of AFS, we applied it to four tasks based on four datasets, including the MNIST dataset, the PathMNIST dataset, the COVIDx dataset, and the ASD dataset, with different data sizes and various architectures of deep learning networks. Our results demonstrate the usability of AFS and its application potential in real-world intelligent healthcare **to enhance privacy protection and data revocation rights. To our knowledge, AFS is the first unified method of auditing and forgetting that could effectively forget the information of the target query dataset from the pre-trained DL model with the guidance of auditing. AFS could generate a smaller model, which requires much less time and GPU memory during the inference, by training with a partial training dataset (~50%) with our novel KP approach.**

3.6. Major concerns #3

RC: *To better understand AFS's applicability and effectiveness in real-world production environments, scaling it to larger models and more data is essential.*

AR: We would like to express our gratitude to the reviewer for the valuable feedback on our research. We agree that it is important to scale AFS to larger models and data sets to fully understand its applicability and effectiveness in real-world production environments. Though we are unable to perform experiments to scale AFS to the current famous large models such as LLM (large language model, eg. chatGPT), we still have tried our best to add a new experiment to address the reviewer's concerns.

We have created an augmented MNIST dataset with 180,000 samples (three times the original MNIST

dataset, 18 times the training dataset we used in the previous manuscript) and used a much deeper MLP with 129,105,930 parameters (around 192 times the original model) as the teacher model and 12,615,690 parameters (around 81 times the original model) as the student model for the 10-class classification task. We also prepared a query dataset with 5,000 samples. Based on the results in Response Table 8, we can prove that it is feasible to apply AFS to deeper neural networks with more data as it demonstrates the ability to simultaneously forget information of the query data and transfer the remaining information from the original model.

Methods	QF_{5000}	Accuracy	F1-score
Independent teacher	1	0.9785	0.9951
Independent student	1	0.9769	0.9942
Independent teacher (k=0.5)	1.27e-24	0.9701	0.9921
Independent student (k=0.5)	1.69e-26	0.9684	0.9903
AFS w/o Audit (k=0.5)	6.33e-12	0.9743	0.9938
AFS (k=0.5)	4.73e-41	0.9707	0.9929

Table 8: Comparison of AFS with other methods on forgetting QF and model performance with the Augment-edMNIST dataset. We present the p-values of auditing models trained with different methods on QF_{5000} and the model performance including the accuracy and F1-score.

3.7. Major concerns #4

RC: *Although the authors focus on four classification tasks, it would be interesting to explore the idea of expanding AFS to regression tasks, which may present challenges in incorporating additional auditing metrics.*

AR: We would like to express our utmost gratitude to the reviewer for providing valuable feedback on our manuscript. The reviewer’s suggestions and comments have been extremely helpful in shaping our work. We agree that extending AFS to regression tasks is a fascinating idea. However, we have encountered some practical limitations that prevent us from pursuing it further. Therefore, we regret to inform the reviewer that we are unable to implement this suggestion due to the following two reasons.

1. The design of auditing doesn’t work for regression tasks.

As mentioned in the manuscript, the auditing process needs to calculate three metrics, including correctness, confidence, and entropy to further calculate the p-value, which will be used in the forgetting process as below.

We must clarify that the calculation of correctness, confidence, and entropy is only applicable to classification tasks, where we have soft logits and hard labels. In regression tasks, we have no theoretical support and can not calculate these metrics, therefore, the auditing component, which is a critical part of AFS, cannot be applied.

EMA is designed as a 2-step process. In the first step, the best threshold for each metric is selected to optimize $(TPR(t) + TNR(t))/2$ based on the calibration dataset as shown in Algorithm 1. Once the thresholds for all metrics are selected, the membership of each sample in the query dataset will be confirmed as at least one metric is larger than the corresponding threshold. In total, three metrics, including correctness, confidence, and entropy, were adopted to further calculate the p-value, which is the key audit metric in AFS as proposed in the previous work. The details for calculating correctness, confidence, and entropy are as below:

Correctness: The target model is trained to predict correctly on training data and may not generalize well on test data. Thus we can define the correctness as:

$$I_{correctness}(F, (\mathbf{x}, y)) = 1 \{ \operatorname{argmax}_i F(\mathbf{x})_i = y \} \quad (4)$$

where F is the deep learning model, \mathbf{x} is the input data, $F(\mathbf{x})$ is the output logits, y is the label, and 1 is the indicator function.

Confidence: The target model is usually more confident in predictions on training data, but less confident in test data. Thus we can define confidence as:

$$I_{confidence}(F, (\mathbf{x}, y)) = 1 \{ F(\mathbf{x})_y \geq \tau_y \} \quad (5)$$

where F is the deep learning model, \mathbf{x} is the input data, $F(\mathbf{x})$ is the output logits, y is the label, $F(\mathbf{x})_y$ is the output logits for label y , τ_y is a threshold for the logit for label y , and 1 is the indicator function.

Entropy: The target model is trained by minimizing the prediction loss over training data and usually has a larger prediction entropy on a test sample. Thus we can define entropy as:

$$I_{entropy}(F, (\mathbf{x}, y)) = 1 \left\{ - \sum_i F(\mathbf{x})_i \log(F(\mathbf{x})_i) \leq \hat{\tau}_y \right\} \quad (6)$$

where F is the deep learning model, \mathbf{x} is the input data, $F(\mathbf{x})$ is the output logits, y is the label, $F(\mathbf{x})_i$

is the output logits for class i , τ_y is a threshold, and 1 is the indicator function.

Once the membership of all samples in the query dataset is confirmed in the previous step, the query dataset will be further evaluated to determine whether the query dataset has been used to train the target pre-trained DL model. A two-sample statistical test is adopted to evaluate the query dataset based on the sample-wise membership and an all-one vector. The p-value of the two-sample statistical test is used as the output of auditing. Given a user-defined threshold α , if $p < \alpha$, then users could conclude that the query dataset was not used for training the target DL model. EMA was re-implemented and integrated into AFS to allow easy and fast auditing.

2. Knowledge purification is limited to classification tasks.

The current version of the Knowledge Purification algorithm requires soft logits in classification tasks to calculate the loss, which makes it difficult to be applied to regression tasks. Additionally, Knowledge Distillation is not specifically designed for regression tasks, which involve predicting continuous values. As noted in the paper "BAM! Born-Again Multi-Task Networks for Natural Language Understanding" by Stanford University and Google Brain, distillation may not work well for regression tasks as there is no distribution over classes passed on by the teacher to aid learning.

While we acknowledge the potential interest of applying AFS to regression tasks, it requires a redesign of both the auditing and knowledge purification algorithms, which is beyond the scope of our current work. Nonetheless, we deeply appreciate the reviewer's efforts and will try our best to take the reviewer's feedback into consideration in our future research endeavours.

3.8. Conclusion

RC: *Despite the aforementioned limitations, this reviewer still believes that the proposed AFS method contributes significantly to enhancing privacy protection and data revocation rights. The paper can be publishable if the listed limitations are addressed.*

AR: Thank the reviewer for the positive assessment of our method. We appreciate the reviewer's feedback and are glad that the reviewer finds our work valuable in enhancing privacy protection and data revocation rights. Hope that our point-to-point response could address the reviewer's concerns.

REVIEWER COMMENTS

Reviewer #2 (Remarks to the Author):

Thanks for the update, revisions and well-detailed response letter.

1- EMA vs AFS: Reviewer would like to note that AFS improvement is not algorithmic but computational through parallelism. The reviewer notes that this reduces the significance of the contribution. The reviewer advises authors to cover algorithmic changes necessary for parallelism as the contribution point.

2- Figure 1 of the response letter: The reviewer advises authors to highlight the significance of this several-second of time difference. E.g., 1 milisec can be life-threatening in real-time systems like autonomous vehicles but means nothing when loading a page in a browser.

3- Authors state that "We have shown that Audit and Forget are algorithmically linked, and our approach provides a novel solution to this problem." Reviewers would agree if achieving such a chaining requires solving a significant problem rather than running one algorithm after another. Could authors clarify the difficulty and significance of conducting Audit and Forget, combined?

4- The reviewer understands the author's point that subsequent teacher students training can be done in a way to keep the model at a similar size. However, what is the requirement for the model size reduction in the first iteration? Please clarify why the model needs to be distilled to a smaller size. Saying that we successfully forget, and as a side effect, we get a smaller model which fits in smaller memory does not sound to be a good motivation.

5- Authors state that "but we believe that our unique research question and methodology justify its contribution to the field" Please elaborate further to show that this is sufficient for a prestigious venue.

6- Table 1 suggests an accuracy drop in subsequent forgetting rounds (from 0.96 to 0.92 in the fifth round). What is the budget here? How do authors model the budget? Where exactly should the organization stop forgetting and re-train the model with data minus the forgotten records?

7- Thanks. The reviewer acknowledges the additional comparisons and the significance of the work done to achieve these comparisons.

8- The reviewer acknowledges the clarifications around metrics and how they are used to obtain p-value comparison.

9- "large companies usually have millions of users or even trillions of users." The reviewer highlights that there are approximately 8 billion people living on the Earth.

Reviewer #3 (Remarks to the Author):

The revision addressed some of the raised issues. But there are still remaining issues.

1 Generalization of the proposed method to regression task.

The paper says that the auditing process needs to calculate three metrics, correctness, confidence, and entropy, which cannot support the regression tasks. There are some confusions here and hope the authors can clarify. The computation of correctness is decided by the indicator functions, output logits, and label. Suppose that we take F as the output, y as the true value, and the indicator function as the loss function, is it possible to extend the proposed method? Correspondingly, the computation of confidence takes a similar strategy and has a threshold for error of $(\text{output}-\text{true value})$. Entropy can be regarded as one of the loss functions.

2. Scalability

MNIST may not be viewed as a large dataset. Although the proposed method might work well for small-sized datasets, it is still not clear how it performs on very large datasets or complex models in terms of its effectiveness and efficiency, and hence further exploration is needed.

Authors' Response to Reviews of NCOMMS-23-05640A-Z

Audit to Forget: A Unified Method to Revoke Patients' Private Data in Intelligent Healthcare

Juexiao Zhou^{1,2,#}, Haoyang Li^{1,2,#}, Xingyu Liao^{1,2}, Bin Zhang^{1,2}, Wenjia He^{1,2}, Zhongxiao Li^{1,2}, Longxi Zhou^{1,2}, Xin Gao^{1,2,*}

¹Computer Science Program, Computer, Electrical and Mathematical Sciences and Engineering Division, King Abdullah University of Science and Technology (KAUST), Thuwal 23955-6900, Kingdom of Saudi Arabia

²Computational Bioscience Research Center, King Abdullah University of Science and Technology, Thuwal 23955-6900, Kingdom of Saudi Arabia

*Corresponding author

#Equal contribution

RC: *Reviewers' Comment*, AR: Authors' Response, □ Manuscript Text

1. General Response

AR: We really appreciate the editor [redacted] and both reviewers' valuable feedback and suggestions, which have extremely helped us improve the quality and clarity of our work. Once again, we would like to express our sincere gratitude. We are pleased to see that we have addressed the majority of the concerns raised by both reviewers in the previous revision. Still, we also acknowledge that there is scope for us to further improve the manuscript at this stage. We are delighted to have the opportunity to elevate the manuscript to the next level with the assistance and guidance of our esteemed reviewers. Therefore, we carefully considered all comments and made necessary changes to further improve the quality and impact of our work. We believe that we have addressed the concerns raised by the reviewers to the best of our ability by either adding new results or clarification. We showed the revised places in the **previous revision** with blue underline and highlighted the **newly edited places** in the manuscript with a **yellow highlight** to make the revised portions of the manuscript clear to the reviewers.

2. Response to Reviewer #2's Comments

RC: *Thanks for the update, revisions and well-detailed response letter.*

AR: We would like to express our heartfelt gratitude to the reviewer for dedicating valuable time to review our manuscript and for providing us with precious and constructive feedback. We recognize the importance of such feedback and sincerely appreciate the reviewer's thoughtful comments and suggestions.

RC: *1- EMA vs AFS: Reviewer would like to note that AFS improvement is not algorithmic but computational through parallelism. The reviewer notes that this reduces the significance of the contribution. The reviewer advises authors to cover algorithmic changes necessary for parallelism as the contribution point.*

AR: We appreciate the reviewer's comment regarding the difference between the Audit module in AFS and EMA. We acknowledge the importance of clarifying this point to highlight the contribution of our work and to avoid overstating contributions.

In general, AFS consists of two modules, the first part is the Audit module developed on EMA and the second part is the Forget module (our knowledge purification algorithm). Though the Audit module in AFS is developed on EMA, we have optimized the computation time by adding parallel computation to the code to achieve faster speed. At this level, we agree that our contribution is computational. However, to implement it, we have also made modifications to the algorithm, as shown in Response Algorithm 1 (Algorithm 1 in the main text), and used some tricks in coding, such as using GPU accelerated matrix computation, so we hope the reviewer could agree that there is a degree of contribution to the algorithm as well. Though the contribution to the Audit module of AFS is not as significant as our knowledge purification algorithm, the improved Audit indeed solves practical problems for acceleration and we will further specify this point in the response to question 2.

We have updated the algorithm 1 in the main text as below:

Algorithm 1 Infer thresholds

Require: The calibration dataset D_{cal} , the pre-trained DL model A and n different metrics (m_1, \dots, m_n) for membership testing.

1: **procedure**

2: Split D_{cal} into training dataset D_{cal}^{train} and test dataset D_{cal}^{test}

3: The calibration model is trained as $f_{D_{cal}} \leftarrow A(D_{cal}^{train})$

4: **for** $m_i \in \{m_1, \dots, m_n\}$ **do in parallel**

5: Compute metrics for training dataset as $M_{train} \leftarrow \{m_i(f_{D_{cal},s} | s \in D_{cal}^{train})\}$

6: Compute metrics for test dataset as $M_{test} \leftarrow \{m_i(f_{D_{cal},s} | s \in D_{cal}^{test})\}$

7: Find $t_i \in \operatorname{argmax}_{t \in [M_{train}, M_{test}]} (\frac{TPR(t) + 2 \cdot TNR(t)}{2})$, where $TPR(t) = \sum_{s \in D_{cal}^{train}} 1_{\{m_i(s) \geq t\}} / |D_{cal}^{train}|$ and $TNR(t) = \sum_{s \in D_{cal}^{test}} 1_{\{m_i(s) < t\}} / |D_{cal}^{test}|$

return The thresholds t_1, \dots, t_n for n metrics

RC: 2- *Figure 1 of the response letter: The reviewer advises authors to highlight the significance of this several-second of time difference. E.g., 1 milisec can be life-threatening in real-time systems like autonomous vehicles but means nothing when loading a page in a browser.*

AR: We acknowledge the reviewer’s concerns regarding the significance of acceleration at the second level when comparing the Audit module of AFS with EMA. We apologize for not explaining this aspect in our previous response.

As shown in Figure 1 (the same one from the previous response letter), the time difference indicated refers to a single epoch in the training process. We explicitly mentioned in the manuscript that the Audit module is referenced during each training epoch to facilitate information forgetting through knowledge purification. To forget, the model is trained with knowledge purification until convergence is reached. The total number of epochs required for convergence depends on factors such as the size of the data and the speed at which the model converges. For example, we need more than 1000 epochs for the MNIST case in the manuscript.

Let’s suppose that we need 1000 epochs for a query dataset of 50,000 samples as shown in Figure 1, and considering the median time of 25 seconds for the Audit module in AFS and 12 seconds for EMA for single auditing, the total time difference would be approximately $1000 * (25 - 12) = 13,000$ seconds (around 3.6 hours). This difference in time is indeed significant, and the gap could be even larger when more epochs are needed or when multiple query datasets need to be forgotten.

The acceleration demonstrated by the Audit module of AFS becomes particularly attractive for companies dealing with larger-scale data and larger deep learning models. Longer GPU computation times mean

increased costs, making the ability to accelerate the forgetting process highly beneficial. By reducing the time required for forgetting, AFS offers potential cost savings for companies by minimizing GPU usage and, subsequently, the associated expenses.

We have updated the main text on Page 3 as below:

To audit the membership of the query dataset, AFS takes a pre-trained DL model and the query dataset as inputs, and determines whether the query dataset has been used for training the target DL model. This function was re-implemented based on EMA, a published MIA-based method to evaluate the membership of a query dataset. Our re-implementation allows quicker and easier usage of auditing by introducing parallel computing in each epoch, which suggests a significant acceleration for a complete forgetting process and could be more attractive to institutions to forget larger-scale data. (Section 2.4 and Supplementary Figure 1).

Figure 1: Audit time of AFS and EMA for different sizes of the query dataset. The y-axis represents time in seconds.

RC: 3- Authors state that "We have shown that Audit and Forget are algorithmically linked, and our approach provides a novel solution to this problem." Reviewers would agree if achieving such a chaining requires solving a significant problem rather than running one algorithm after another. Could authors clarify the difficulty and significance of conducting Audit and Forget, combined?

AR: We greatly value the reviewer's feedback regarding the challenges and importance of combining Audit and Forget processes. We acknowledge the concern and recognize the need to provide a more comprehensive explanation of the unique advantages and potential synergies that arise from this combination. We will elaborate on these points below:

First, without combining Audit and Forget algorithmically, the processes of Audit and Forget are conducted as separate steps. For example, to forget a query dataset from a pre-trained model, the following steps are typically performed:

1. Determine if the data is remembered by the model by calculating relevant metrics such as Audit.
2. Employ specific algorithms (e.g., SISA, EU-k, CF-k as mentioned in our manuscript) to generate new models. However, these methods do not leverage the knowledge of whether the data is remembered by the model, resulting in uncertain performance in the forgetting process.
3. Assess whether the data has been successfully forgotten by examining multiple models generated in the previous step. If not, we repeat the previous process.
4. Finally we choose the model with the best forgetting performance as the new model.

One limitation of the previous procedures is that the information regarding whether the data is remembered by the model is not fully utilized during the training phase, but only considered during the model selection phase. Thus the forgetting performance can be further improved as we could make full use of the information of "if the data is remembered by the model" in the training phase. By incorporating the knowledge of whether the data is remembered by the model as one of the goal functions, we can enhance the directed forgetting process and improve the overall forgetting performance.

To achieve that, we integrate the information of "if the data is remembered by the model" into the forgetting process, specifically in the knowledge purification stage. By doing so, we utilize this information to update the model during each step of forgetting, establishing an algorithmic link between the information and the model training process. The benefits of doing so are also demonstrated by comparing AFS to other methods in the manuscript. Our results clearly indicate that incorporating this information into the training process is more effective for forgetting than solely relying on it for model selection.

To summarize, when Audit and Forget are executed as separate steps, the information regarding whether the data is remembered by the model is not fully utilized. In contrast, our approach establishes an algorithmic connection between Audit and Forget, enabling forgetting to occur during the training process itself. Linking the two algorithmically is not technically difficult in itself, but proposing this idea is innovative and effective.

RC: 4- *The reviewer understands the author’s point that subsequent teacher students training can be done in a way to keep the model at a similar size. However, what is the requirement for the model size reduction in the first iteration? Please clarify why the model needs to be distilled to a smaller size. Saying that we successfully forget, and as a side effect, we get a smaller model which fits in smaller memory does not sound to be a good motivation.*

AR: We would like to express our gratitude to the reviewer for their valuable feedback. We highly appreciate the comprehensive evaluation and acknowledge the concern raised regarding the motivation for reducing the model size in the initial iteration.

Model compression is a widely utilized technique that holds significant potential for various real-life applications. One common approach to model compression is knowledge distillation, which aims to transfer knowledge from a larger, more complex model to a smaller one. Our knowledge purification could also be a technique of model compression when we need to compress the model.

Firstly, we answer the question of “why the model needs to be distilled to a smaller size”. The need to distil or compress a model to a smaller size arises from practical considerations during the model deployment phase. While larger models often exhibit better learning capabilities and performance during the training phase, deploying these larger models on other devices or providing cloud-based inference services can introduce challenges related to computational resources and inference time. Therefore, during the model training phase, we tend to use larger models. But in the model inference phase, we prefer to use a small model. Model compression techniques offer valuable solutions by reducing the model size, thereby saving computational resources and reducing inference time. This compression process involves transforming the model from its larger, resource-intensive training phase to a smaller, more streamlined form suitable for deployment.

For example, some current large language models have an enormous number of parameters, such as the 176B BLOOM model that requires more than one hundred A100 GPUs to train and at least 6 A100 GPUs to host. This poses challenges for most users who lack the necessary resources to deploy these models locally. Disregarding the most popular large language models, even the most common CNN models in medical diagnostics face the same problem (to reduce inference time from hours or minutes to seconds). To address this issue, extensive research has been conducted on leveraging knowledge distillation techniques to compress these large models without compromising their effectiveness. The aim is to create smaller models that can

run efficiently with reduced computational resources and faster inference times. In the medical field, in particular, faster inference models are of greater value as the reviewer stated previously “1 milisec could be life-threatening in the real-time system”. Therefore, generating a smaller model which fits in smaller memory and faster inference time indeed is a good motivation in the community. Meanwhile, previous studies have also demonstrated that training a small model directly may not be as effective as training a large model and subsequently compressing it to a smaller size. Hence, the practice of distilling models to a smaller size is prevalent in various real-life scenarios.

In addition to model compression, our knowledge distillation approach introduces an additional requirement: the ability to forget specific information during the knowledge transfer process. As mentioned in the previous response, the KP algorithm has two application scenarios. The first scenario involves forgetting knowledge while compressing the model, while the second scenario focuses on achieving knowledge forgetting without model compression. To provide further clarification on this point, we have included Figure 2D (Figure 5D in the main text) as below:

Figure 2: Performance of forgetting using AFS on four datasets. **A.** The p-value of auditing on a small query dataset and a large query dataset (QF) and the accuracy of models trained with different methods, including Original (Independent teacher trained with the complete training dataset), SISA, Data Deletion (the Independent student model trained with partial training dataset and $k = 0.5$), AFS (w/o Audit) and AFS. **B.** The number of parameters for the original large model and the new small model generated by AFS. **C.** The qualitative evaluation of three methods, including Original (Independent teacher trained with the complete training dataset), SISA, CF-k, EU-k, Data Deletion (the Independent student model trained with partial training dataset and $k = 0.5$), and AFS on five dimensions (Ability to forget, accuracy, size of dataset needed for training, size of the generated model and the efficiency of training). A larger value means a stronger ability to forget, higher model accuracy, a smaller size of dataset needed for training, a smaller size of the generated model, and better efficiency of training. **D.** Illustration of how to incorporate AFS into real-world applications.

Here we showed how to incorporate AFS into real-world applications. The Knowledge Purification (KP) algorithm can be effectively utilized in two distinct scenarios: forgetting prior to model compression and forgetting after obtaining a compressed model, which will be further explained in the response to question 6.

The requirement for model size reduction in the first iteration depends on the specific needs and constraints of the application scenario. Whether model compression is necessary and to what extent it should be applied will vary based on the target platform's computational resource limitations and deployment requirements for the student models. For example, deploying a model requiring 100GB of GPU memory to a platform with only 8GB of GPU memory, we will need a significant level of model compression in the first round. The exact ratio of model compression will depend on the practical limitations and constraints of the target platform. It is important to note that there is no fixed standard or predefined threshold for model compression. Instead, the level of compression needed is determined by the specific circumstances and constraints of the deployment scenario. Factors such as available computational resources, memory limitations, and other practical considerations will guide the decision-making process when determining the appropriate level of model compression in the first iteration.

RC: 5- Authors state that "but we believe that our unique research question and methodology justify its contribution to the field" Please elaborate further to show that this is sufficient for a prestigious venue.

AR: We sincerely appreciate the reviewer's feedback on the contribution of our method and the potential significance of knowledge purification as a research topic. Hope our responses to questions 3 and 4 and also in the response to question 6 later could address this concern, we will provide a brief summary here.

Firstly, our approach leverages the information of whether the data is remembered by the model to achieve targeted forgetting algorithmically, resulting in improved forgetting performance. By incorporating this information into the forgetting process, we can effectively forget specific data from trained models. Additionally, as the field of machine learning continues to advance, privacy concerns have become increasingly important. Forgetting specific data from trained models has become a common and vital requirement, particularly in domains such as the medical field. Protecting the right of users to revoke their private data is crucial, and the concept of knowledge purification holds significant potential in addressing this need. Considering these factors, knowledge purification as a novel concept has the potential to garner widespread attention and become a subject of extensive research.

RC: 6- Table 1 suggests an accuracy drop in subsequent forgetting rounds (from 0.96 to 0.92 in the fifth round). What is the budget here? How do authors model the budget? Where exactly should the organization stop forgetting and re-train the model with data minus the forgotten records?

AR: We would like to thank the reviewer for raising the insightful question regarding the budget on accuracy drop and a potential guideline for when to stop forgetting and train a new model. This question is extremely valuable and helpful. It is very important to further elaborate this concern.

Indeed, it is widely acknowledged that while users may request institutions to forget their data, institutions themselves are constantly collecting new data for the purpose of training new models. This is particularly prevalent in various industries where the continuous acquisition of new data is essential for training state-of-the-art models and maintaining a competitive edge. Therefore, we added a new figure to the manuscript as shown in Response Figure 2 (Figure 5D in the main text) to further explain this point. The figure illustrates the workflow of the institution in two distinct phases. In the first phase, the institution collects an initial dataset and trains a large model during the training phase. Following the training phase, the model may need to be compressed to meet specific requirements for deployment. As previously mentioned, the compression ratio of the model depends on the actual demand and can vary accordingly.

Patients' requests for data forgetting may occur in two different phases. The first phase relates to requests made before model compression, where patients request the institution to forget their data from the initial dataset. The second phase pertains to requests made after model compression, where patients seek to have their data forgotten from the compressed model. In both cases mentioned, the AFS algorithm can be employed to forget information from the model.

In the first case, AFS facilitates model compression while performing forgetting. This ensures that the compressed model not only meets the requirements for deployment but also respects data privacy by removing unnecessary or sensitive information. In the second case, AFS is utilized to forget information from the model without involving model compression. This allows the institution to respond to data forget requests while retaining the compressed model's structure and size. If additional forget requests arise (as referred to as continuous forgetting), the AFS algorithm can be repeatedly applied to forget the relevant information. However, it is crucial to note that this process cannot continue indefinitely. Each forgetting step leads to a certain loss of model effectiveness, and the AFS algorithm relies on a certain percentage of raw data (e.g., $k=0.5$) to facilitate information transfer.

As pointed out by the reviewer, "Table 1 suggests an accuracy drop in subsequent forgetting rounds (from 0.96 to 0.92 in the fifth round)", which happens when we have 5 consecutive rounds, each time forgetting 5% of the data. However, it is important to consider the scale of forgetting requests in real-life scenarios. In practical situations, organizations may have vast amounts of data, and a 5% forgetting request is already substantial and is not expected to occur frequently. In contrast, we would typically anticipate much smaller scale forgetting requests. For example, a forgetting request of 0.1% or even smaller may be more common.

With such smaller-scale forgetting requests, the negative impact on the model is relatively small as well. The model's performance is less affected, allowing organizations to accommodate forget requests without significant accuracy degradation.

Indeed, the relationship between the decline in accuracy and the amount of forgotten data is not strictly linear. This is due to the fact that, during the forgetting process, we still require a certain percentage of training data to facilitate the transfer of retained information. The accuracy decline is influenced by various factors, including the unique information contained within the data to be forgotten and the degree of similarity between the data to be forgotten and the retained data.

Therefore, to answer the question "What is the budget here?", we may need to specify it according to the actual problem. We understand that budget means how much of the model performance degradation we can accept when we need to balance privacy and model effectiveness. The modelling of the budget should be case-specific. Thus, the selection of budget needs to refer to the actual situation. For example, in critical medical diagnosis applications, even a small decrease in accuracy, such as 0.5%, could have significant consequences, potentially putting patients at risk if incorrect diagnoses are made. In such cases, maintaining a high level of model performance is crucial, and a stricter budget on accuracy degradation may be necessary to prioritize patient safety. On the other hand, in more general scenarios where some degree of fault tolerance is permissible, such as a preliminary diagnosis, organizations may be more willing to accept a certain percentage of performance decrease in order to protect the users' right to revoke their data and ensure data privacy.

Based on the response above, to answer "Where exactly should the organization stop forgetting and re-train the model with data minus the forgotten records?", two possible scenarios can be considered. The first scenario occurs when the number of forgetting requests becomes too high, leading to a significant degradation in the current model's performance that exceeds the institution's predefined budget. In such cases, it may be necessary to stop the forgetting process and initiate the retraining of a new model. The second scenario arises when the institution introduces new data into the system. In this case, instead of continuing to forget specific records from the existing model, the institution can incorporate the retained data along with the new data to train a new model as shown in 2 (Main Figure 5D).

We have updated the discussion section of the main text as below:

In practice, the size of the student model could be manually adjusted when applying AFS to meet specific requirements. In the initial stages, we could make the student model smaller than the teacher model to achieve model compression while forgetting private information. However, instead of continuously generating smaller and smaller student models as knowledge is forgotten, we could maintain the

size of the student model once it reaches a certain threshold (e.g., small enough to meet our needs). In doing so, AFS would focus solely on forgetting knowledge and transferring the remaining knowledge, without the need for further model compression.

AFS could be incorporated into the workflow of institutions as shown in Figure 2D. Patient's requests for data forgetting may occur in two different phases. The first phase relates to requests made before model compression, where patients request the institution to forget their data from the initial dataset. The second phase pertains to requests made after model compression, where patients seek to have their data forgotten from the compressed model. In both cases, AFS could be employed to forget information from the model. In the first case, AFS facilitates model compression while performing forgetting, which ensures that the compressed model not only meets the requirements for deployment but also respects data privacy by removing sensitive information. In the second case, AFS could be utilized to forget information from the model without involving model compression. This allows the institution to respond to data forget requests while retaining the compressed model's structure and size. The institution may stop forgetting and retrain a new model in two possible scenarios. The first scenario occurs when the number of forgetting requests becomes too high, leading to a significant degradation in the current model's performance that exceeds the institution's predefined budget. In such cases, it may be necessary to stop the forgetting process and initiate the retraining of a new model. The second scenario arises when the institution introduces new data into the system. In this case, instead of continuing to forget specific records from the existing model, the institution can incorporate the retained data along with the new data to train a new model.

RC: *7- Thanks. The reviewer acknowledges the additional comparisons and the significance of the work done to achieve these comparisons.*

AR: We greatly appreciate the reviewers' valuable time and constructive suggestions. The feedback has been instrumental in enhancing the quality and clarity of our work.

RC: *8- The reviewer acknowledges the clarifications around metrics and how they are used to obtain p-value comparison.*

AR: We are very grateful for the reviewer's suggestion to clarify the manuscript.

RC: *9- "large companies usually have millions of users or even trillions of users." The reviewer highlights that there are approximately 8 billion people living on the Earth.*

AR: Thank the reviewer for bringing this up, and we apologize that we misrepresented it in the previous response. What we were trying to say was "large companies usually have millions of users or even trillions of samples

from users". In practical scenarios, a single user may provide several samples, which collectively contribute to the large-scale dataset used for training models.

3. Response to Reviewer #3's Comments

RC: *The revision addressed some of the raised issues. But there are still remaining issues.*

AR: We would like to express our sincere appreciation to the reviewer for taking the valuable time to review our manuscript and providing us with constructive feedback. We value the opportunity to improve our manuscript once again and elevate our manuscript to the next level with the assistance and guidance of the esteemed reviewer.

RC: *1 Generalization of the proposed method to regression task. The paper says that the auditing process needs to calculate three metrics, correctness, confidence, and entropy, which cannot support the regression tasks. There are some confusions here and hope the authors can clarify. The computation of correctness is decided by the indicator functions, output logits, and label. Suppose that we take F as the output, y as the true value, and the indicator function as the loss function, is it possible to extend the proposed method? Correspondingly, the computation of confidence takes a similar strategy and has a threshold for error of (output-true value). Entropy can be regarded as one of the loss functions.*

AR: We would like to express our gratitude to the reviewer for the valuable suggestion. We appreciate the feedback on incorporating correctness, confidence, and entropy directly into the loss function to extend the applicability of our method to regression tasks. To further elaborate on this point, we will show the practical difficulties in detail below:

Let's suppose \mathbf{X} to be all samples, \mathbf{x} to be one input sample, $F(\mathbf{x})$ to be the predicted vector and \mathbf{y} to be the label vector (e.g. one-hot encoded label) and y is the label.

So, for classification tasks (let's suppose 5 classes, labelled as 0-4), let's look at a sample \mathbf{x} from \mathbf{X} and suppose the prediction of \mathbf{x} to be [0.1, 0.1, 0.6, 0.1, 0.1] the label \mathbf{y} of \mathbf{x} to be [0, 0, 1, 0, 0] and label y to be 2.

However, for the regression task, let us look at a sample x again from \mathbf{x} . Let's suppose the prediction of \mathbf{x} to be [7.5], the label \mathbf{y} to be [10.3] and the label y to be 10.3.

Correctness

As we defined in the manuscript, correctness is

$$I_{correctness}(F, (\mathbf{x}, y)) = 1\{\arg\max_i F(\mathbf{x})_i = y\} \quad (1)$$

For the classification task, with $1\{\arg\max_i F(\mathbf{x})_i = y\}$, 1 means the label y of \mathbf{x} has the highest prediction score.

For the regression task, we can not state it the same way. Still, as suggested by the reviewer, we can modify the definition to:

$$I_{correctness}(F, (\mathbf{x}, y)) = 1\{F(\mathbf{x}) = y\} \quad (2)$$

where $1\{F(\mathbf{x}) = y\}$ means if the regression value is exactly the same as the label. If so, we say it is correct.

We appreciate the reviewer's insight regarding the definition of correctness in the context of regression problems. While predicting the exact same value as the label in regression tasks can be challenging, we acknowledge that a strict definition of correctness may lead to most samples being classified as 0 in terms of correctness. Therefore, correctness is not a good metric for regression tasks. Still, by incorporating the definition of confidence below, as suggested by the reviewer, we can potentially address this issue. The introduction of an "error" term, which quantifies the deviation between the predicted value and the actual label, can provide a measure of confidence in the prediction.

Confidence

Similarly, as we defined in the manuscript, confidence is

$$I_{confidence}(F, (\mathbf{x}, y)) = 1\{F(\mathbf{x})_y \geq \tau_y\} \quad (3)$$

For the classification task, with $1\{F(\mathbf{x})_y \geq \tau_y\}$, 1 means the prediction score for label y of \mathbf{x} is larger than the threshold τ_y we defined from the calibration dataset.

For the regression task, with $1\{F(\mathbf{x})_y \geq \tau_y\}$, 1 means the regression value of \mathbf{x} is larger than the threshold τ_y we defined from the calibration dataset. Obviously, we can not state it the same way. Still, as suggested by the reviewer, we can calculate the error between the prediction score and the label, and modify the definition to:

$$I_{confidence}(F, (\mathbf{x}, y)) = 1\{F(\mathbf{x})_y - y \geq \tau_y\} \quad (4)$$

where with $1\{F(\mathbf{x})_y - y \geq \tau_y\}$, 1 means the regression value of \mathbf{x} minus the label y (the error) is larger than the threshold τ_y we defined from the calibration dataset. Now, we need to see what is τ_y . However, as shown in Response Algorithm 2 (Algorithm 1 in the main text), the definition of τ_y does not work for the regression task, because we are unable to perform step 7. Thus we need to redesign the whole algorithm for inferring the thresholds.

As suggested by the reviewer, we could potentially re-design line 7 of the algorithm as Algorithm 3:

Algorithm 2 Infer thresholds

Require: The calibration dataset D_{cal} , the pre-trained DL model A and n different metrics (m_1, \dots, m_n) for membership testing.

1: procedure

2: Split D_{cal} into training dataset D_{cal}^{train} and test dataset D_{cal}^{test}

3: The calibration model is trained as $f_{D_{cal}} \leftarrow A(D_{cal}^{train})$

4: **for** $m_i \in \{m_1, \dots, m_n\}$ **do** in parallel

5: Compute metrics for training dataset as $M_{train} \leftarrow \{m_i(f_{D_{cal},s} | s \in D_{cal}^{train})\}$

6: Compute metrics for test dataset as $M_{test} \leftarrow \{m_i(f_{D_{cal},s} | s \in D_{cal}^{test})\}$

$\{m_i(f_D$

7: Find $t_i \in \operatorname{argmax}_{t \in [M_{train}, M_{test}]} (\frac{TPR(t) + 2NR(t)}{cal})$, where $TPR(t) = \sum_{s \in D_{cal}^{train}} \mathbf{1}\{m_i(s) \geq$

$\frac{t}{|D_{cal}^{train}|}$ and $NR(t) = \sum_{s \in D_{cal}^{test}} \mathbf{1}\{m_i(s) < t\} / |D_{cal}^{test}|$

return The thresholds t_1, \dots, t_n for n metrics

Algorithm 3 Infer thresholds (Regression)

Require: The calibration dataset D_{cal} , the pre-trained DL model A and n different metrics (m_1, \dots, m_n) for membership testing.

1: procedure

2: Split D_{cal} into training dataset D_{cal}^{train} and test dataset D_{cal}^{test}

3: The calibration model is trained as $f_{D_{cal}} \leftarrow A(D_{cal}^{train})$

4: **for** $m_i \in \{m_1, \dots, m_n\}$ **do** in parallel

5: Compute metrics for training dataset as $M_{train} \leftarrow \{m_i(f_{D_{cal},s} | s \in D_{cal}^{train})\}$

6: Compute metrics for test dataset as $M_{test} \leftarrow \{m_i(f_{D_{cal},s} | s \in D_{cal}^{test})\}$

$\{m_i(f_D$

7: Find $t_i \in \operatorname{argmax}_{t \in [M_{train}, M_{test}]} (\frac{TPR(t) + 2NR(t)}{cal})$, where $TPR(t) = \sum_{s \in D_{cal}^{train}} \mathbf{1}\{m_i(s) -$

$\frac{\sum_{y_s \leq t} y_s}{|D_{cal}^{train}|}$ and $NR(t) = \sum_{s \in D_{cal}^{test}} \mathbf{1}\{m_i(s) - y_s > t\} / |D_{cal}^{test}|$

return

Entropy

Similarly, as we defined in the manuscript, entropy is

$$I_{entropy}(F, (\mathbf{x}, y)) = 1 \left\{ - \sum_i F(\mathbf{x})_i \log(F(\mathbf{x})_i) \leq \hat{\tau}_y \right\} \quad (5)$$

So, for classification tasks (let's suppose 5 classes, labelled as 0-4), let's look at a sample \mathbf{x} from \mathbf{X} and suppose the prediction of \mathbf{x} to be [0.1, 0.1, 0.6, 0.1, 0.1] the label \mathbf{y} of x to be [0, 0, 1, 0, 0] and label y to be 2. Calculating the entropy across labels as $-\sum_i F(\mathbf{x})_i \log(F(\mathbf{x})_i)$ is meaningful.

However, for the regression task, let us look at a sample x again from \mathbf{x} . Let's suppose the prediction of \mathbf{x} to be [7.5], the label \mathbf{y} to be [10.3] and the label y to be 10.3. Now, calculating the entropy as $-\sum_i F(\mathbf{x})_i \log(F(\mathbf{x})_i)$ is meaningless, because we no longer have information across multiple labels.

Summary

Therefore, the current algorithm design in the manuscript is not applicable to the regression task. Even though we can make some modifications to make the definition potentially work on the regression problem, such as the confidence alone, it is important to note that this approach only addresses a specific aspect (Audit) and does not encompass the entire framework (AFS, Audit and Forget). As we explained in the previous response, finally, we still end up transferring information with knowledge purification. As mentioned in our previous response, the knowledge purification process ultimately involves transferring information from the model. However, the regression task lacks the soft logits representation across multiple labels, which is a crucial component for effective knowledge purification and knowledge distillation. Previous studies have shown the limitations of applying knowledge distillation to regression tasks, even when replacing the loss on soft logits with the mean squared error (MSE) loss (<https://github.com/google-research/google-research/issues/911>). To fully address this issue, it would require the development of a separate framework specifically designed for regression problems, distinct from knowledge distillation or knowledge purification. However, the scope of the current manuscript does not encompass the exploration of such a regression-specific framework.

We appreciate the reviewer's understanding of the limitations and applicability of the current algorithm design in the manuscript for regression tasks. Exploring novel frameworks and approaches specifically tailored to regression problems remains an important avenue for future research.

We also do recognize the limitation of our proposed method in the main text (Page 12, line 677) as below:

Finally, the current version of AFS could not be applicable for regression tasks because the design of auditing metrics does not work for regression tasks and KP is limited to classification tasks only. Despite these limitations, we believe that AFS will make a valuable contribution towards better protection of people's privacy and the right to revoked data with the rapid development of intelligent healthcare.

RC: 2. Scalability. *MNIST may not be viewed as a large dataset. Although the proposed method might work well for small-sized datasets, it is still not clear how it performs on very large datasets or complex models in terms of its effectiveness and efficiency, and hence further exploration is needed.*

AR: Thank the reviewer for the valuable feedback. We understand the concern regarding the size of the dataset and model complexity. We will further explain the experiment in the previous response and add a new experiment with a larger dataset and a more complex deep learning model.

In the previous response, we didn't just use the original MNIST dataset but an augmented MNIST dataset with 180,000 samples, which is three times the original MNIST dataset, and 18 times the training dataset we used in the previous manuscript. We also used a much deeper MLP with 129,105,930 parameters (around 192 times the original model) as the teacher model and 12,615,690 parameters (around 81 times the original model) as the student model for the 10-class classification task. We also prepared a query dataset with 5,000 samples.

Still, we have great respect for the concern of the reviewer. Therefore, to further address the issue, we prepared a much larger dataset named ImageNet that contains 14,197,122 samples, which is 236 times the MNIST dataset. To speed up model training, we used the down-sampled version containing 2,562,320 images with a shape of 64x64 from 1000 classes, which still took us around 3 days to train the base model. We also used a more complex network called VGG-19 with 143,667,240 parameters as the teacher model and VGG-11 with 132,863,336 parameters as the student model for the 1000-class classification task. We also prepared a query dataset with 5,000 samples. Based on the results in Response Table 1, it should be adequate to prove that it is feasible to apply AFS to more complex neural networks and large datasets.

Methods	QF_{5000}	Accuracy	F1-score
Independent teacher	1	0.8421	0.9273
Independent student	1	0.8312	0.8901
Independent teacher (k=0.5)	1.88e-39	0.5056	0.6891
Independent student (k=0.5)	1.52e-57	0.4994	0.6758
AFS (k=0.5)	8.13e-245	0.7903	0.8502

Table 1: Comparison of AFS with other methods on forgetting QF and model performance with the ImageNet dataset. We present the p-values of auditing models trained with different methods on QF_{5000} and the model performance including the accuracy and F1-score.

REVIEWERS' COMMENTS

Reviewer #2 (Remarks to the Author):

I thank the authors for their thorough responses. There is no other concern from my side.

Reviewer #3 (Remarks to the Author):

This reviewer would like to thank the authors for the thorough explanation and the clarification on why the current algorithm is not directly applicable to regression tasks. The absence of a soft logits representation across multiple labels in regression tasks and the reliance of the method on this representation indeed make it challenging to directly apply the proposed approach to such tasks. Nevertheless, it is encouraging to see that the authors intend to investigate new frameworks and strategies specifically designed for regression problems in their future research.

Authors' Response to Reviews of

NCOMMS-23-05640B

Audit to Forget: A Unified Method to Revoke Patients' Private Data in Intelligent Healthcare

Juexiao Zhou^{1,2,#}, Haoyang Li^{1,2,#}, Xingyu Liao^{1,2}, Bin Zhang^{1,2}, Wenjia He^{1,2}, Zhongxiao Li^{1,2}, Longxi Zhou^{1,2}, Xin Gao^{1,2,*}

¹Computer Science Program, Computer, Electrical and Mathematical Sciences and Engineering Division, King Abdullah University of Science and Technology (KAUST), Thuwal 23955-6900, Kingdom of Saudi Arabia

²Computational Bioscience Research Center, King Abdullah University of Science and Technology, Thuwal 23955-6900, Kingdom of Saudi Arabia

*Corresponding author

#Equal contribution

RC: *Reviewers' Comment*, **AR:** Authors' Response, □ Manuscript Text

1. General Response

AR: We really appreciate the editor [redacted] and both reviewers' valuable feedback and suggestions, which have extremely helped us improve the quality and clarity of our work. We are pleased to see that both reviewers are satisfied with the previous revision.

2. Response to Reviewer #2's Comments

RC: *I thank the authors for their thorough responses. There is no other concern from my side.*

AR: We would like to express our heartfelt gratitude to the reviewer for dedicating valuable time to review our manuscript and for providing us with precious and constructive feedback.

3. Response to Reviewer #3's Comments

RC: *This reviewer would like to thank the authors for the thorough explanation and the clarification on why the current algorithm is not directly applicable to regression tasks. The absence of a soft logits representation across multiple labels in regression tasks and the reliance of the method on this representation indeed make it challenging to directly apply the proposed approach to such tasks. Nevertheless, it is encouraging to see that the authors intend to investigate new frameworks and strategies specifically designed for regression problems in their future research.*

AR: We would like to express our sincere appreciation to the reviewer for taking valuable time and providing us with constructive feedback. We are very grateful to the reviewer for pointing out a valuable research direction, and we will focus our future work on how to design new solutions for the regression task.